# DIFFERENTIABLE TOP-$k$: FROM ONE-HOT TO $k$-HOT

## ABSTRACT

The one-hot representation, argmax operator, and its differentiable relaxation, softmax, are ubiquitous in machine learning. These building blocks lie at the heart of everything from the cross-entropy loss and attention mechanism to differentiable sampling. Their $k$-hot counterparts, however, are not as universal. In this paper, we consolidate the literature on differentiable top-$k$, showing how the $k$-capped simplex connects relaxed top-$k$ operators and $\pi$ps sampling to form an intuitive generalization of one-hot sampling. In addition, we propose sigmoid top-$k$, a scalable relaxation of the top-$k$ operator that is fully differentiable and defined for continuous $k$. We validate our approach empirically and demonstrate its computational efficiency.

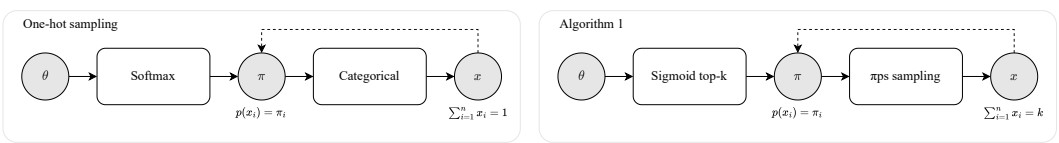

Figure 1: **Effortless $k$-hot sampling.** We propose a method for differentiable $k$-hot sampling as simple as one-hot sampling. The resulting method is modular and computationally efficient.

## 1 INTRODUCTION

**One- and $k$-hot vectors.** A cornerstone of machine learning is the one-hot representation. For numbers 1 through $n$, the corresponding set of **one-hot vectors** is

$$\{0,1\}_1^n := \{\boldsymbol{x} \in \{0,1\}^n \mid \sum_{i=1}^n x_i = 1\}. \tag{1}$$

These vectors model choosing one out of $n$ things. Naturally, they are a useful modeling tool in machine learning with innumerable applications. An important function relating to one-hot vectors is the one-hot **argmax** operator

$$\text{argmax} : \mathbb{R}^n \to \{0,1\}_1^n, \tag{2}$$

that outputs the one-hot vector corresponding to the index of the input's largest value. This argmax should not be confused with the argmax in optimization. One-hot vectors can be generalized by

Table 1: **Overview of top-$k$.** We categorize top-$k$ into mutually exclusive categories: deterministic and stochastic, differentiable and non-differentiable. The table shows four quadrants of $k$-hot problems, along with their one-hot equivalents. Each problem has been addressed separately in previous work. We find that principled approaches to relaxed top-$k$ and $k$-hot sampling can be combined to simplify the problem of differentiable $k$-hot sampling. In doing this, we propose a new relaxed top-$k$ function, sigmoid top-$k$, and review $\pi$ps sampling—a precise approach to $k$-hot sampling that is still largely unknown in machine learning. We also refer to the diagram in Appendix E for an overview.

|  | Deterministic | Stochastic |
|---|---|---|
| Non-differentiable | Top-$k$
Generalized argmax | $k$-hot sampling
Generalized categorical |
| Differentiable | Relaxed top-$k$
Generalized softmax | Differentiable $k$-hot sampling
Differentiable one-hot sampling |

allowing $k$ "hot" indices instead of just one. The set of such **$k$-hot vectors** is

$$\{0,1\}_k^n := \{\boldsymbol{x} \in \{0,1\}^n \mid \textstyle\sum_{i=1}^n x_i = k\}. \tag{3}$$

These vectors model choosing $k$ out of $n$ things without replacement. They can model concepts such as sparsity and subsets. The $k$-hot counterpart to the one-hot argmax is the **top-$k$** operator

$$\text{top-}k : \mathbb{R}^n \to \{0,1\}_k^n, \tag{4}$$

that outputs the $k$-hot vector corresponding to the indices of the input's $k$ largest values.

**Differentiable relaxations.** A challenging aspect of discrete representations like one-hot and $k$-hot vectors is the non-differentiability of argmax and top-$k$. At the same time, gradient-based optimization is currently the predominant approach to optimization in machine learning. This has motivated the development of differentiable relaxations, the most well-known of which is **softmax** that relaxes argmax using the Boltzmann distribution

$$\text{softmax}(\boldsymbol{x}) := \frac{e^{\boldsymbol{x}}}{\sum_{i=1}^n e^{x_i}}. \tag{5}$$

Softmax is widely used in classification networks, attention mechanisms, reinforcement learning, and probabilistic models to parameterize categorical distributions. In §2.1 we discuss how to relax top-$k$ in a similar way.

**Differentiable sampling.** Stochastic relaxation in the form of differentiable sampling is another approach to cope with non-differentiability. For one-hot vectors, this involves drawing a sample from the categorical distribution and using a **gradient estimate**, such as the straight-through (Bengio et al., 2013) or score function estimator (Williams, 1992). In §2.2 we show how $\pi$ps sampling (Tillé, 2006) generalizes the categorical distribution to the $k$-hot case, and in §2.4 we show how popular gradient estimates used for the categorical distribution extend to this approach. Another approach to differentiable sampling is drawing **relaxed samples** from the Gumbel–softmax distribution (Jang et al., 2017; Maddison et al., 2017) and applying the reparametrization trick (Kingma & Welling, 2014). Xie & Ermon (2019) generalized this to the $k$-hot case, building upon Gumbel top-$k$ sampling (Kool et al., 2020). We examine the top-$k$ relaxation used in the original work and improve it with our proposed relaxation in §2.1.

**Applications.** Applications of $k$-hot vectors include beam search (Goyal et al., 2018; Kool et al., 2020), sparse autoencoders (Gao et al., 2025), sparse networks, mixture of experts (Sander et al., 2023), nearest-neighbors (Plötz & Roth, 2018), and more. We note that differentiable applications have been limited by their scalability.

**Contributions.**

- **Generalizing softmax.** We propose sigmoid top-$k$ as a simple and flexible counterpart to softmax for $k$-hot vectors (§2.1). It is computed by solving a scalar root-finding problem, making it scalable to high dimensions. In Proposition 2, we prove that it is equivalent to solving an entropy-regularized optimization problem (Blondel et al., 2019).

- **Generalizing the categorical distribution.** We show how $\pi$ps sampling generalizes sampling to $k$-hot vectors in the sense of sampling with given inclusion probabilities (§2.2). Additionally, we demonstrate how a phantom unit (Bondesson & Grafström, 2011) enables principled learning of $k$ (§2.3).

- **Estimating gradients.** Mirroring differentiable one-hot sampling, we approach differentiable $k$-hot sampling as a composition of relaxed top-$k$ and $\pi$ps sampling. We prove that the expected straight-through gradient is a first-order estimator by extending the proof in Liu et al. (2023) (§2.4). Moreover, we examine previous works from this perspective and identify improvements (§3).

- **Drawing relaxed samples.** We use sigmoid top-$k$ to draw reparameterizable relaxed $k$-hot samples (Xie & Ermon, 2019), improving the time complexity from $\mathcal{O}(nk)$ to $\mathcal{O}(n)$ and producing improved experimental results (§2.5).

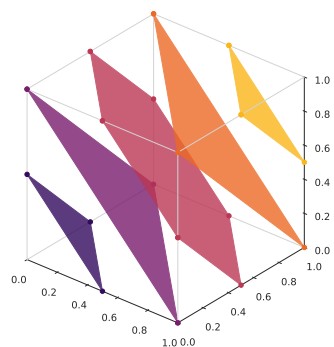 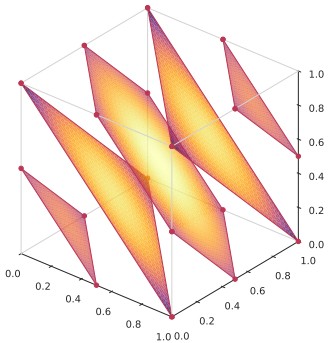

Figure 2: **The $k$-capped simplex**. **Left:** The set of points in the $k$-capped simplex $\Delta_k^{n-1}$ is the intersection of the unit hypercube and a hyperplane, shown here in $n = 3$ dimensions for different values of $k$. The standard probability simplex $\Delta^{n-1}$ corresponds to $k = 1$. **Right:** Elementwise binary entropy over the $k$-capped simplex. This is the regularization term in the optimization problem of Proposition 2.

## 2 METHOD

### 2.1 RELAXED TOP-$k$

**The $k$-capped simplex.** As stated in §1, the softmax function,

$$\text{softmax} : \mathbb{R}^n \to \Delta^{n-1}, \tag{6}$$

serves as a differentiable relaxation of the argmax operator. We want a corresponding relaxation of the top-$k$ operator. Before designing such a function, let us consider softmax's codomain, the **simplex**,

$$\Delta^{n-1} := \{\boldsymbol{\pi} \in [0,1]^n \mid \textstyle\sum_{i=1}^n \pi_i = 1\}. \tag{7}$$

Mirroring how one-hot vectors generalize to $k$-hot vectors, the simplex can be generalized to the **$k$-capped simplex** (Wang & Lu, 2015; Kong et al., 2020; Ang et al., 2021),

$$\Delta_k^{n-1} := \{\boldsymbol{\pi} \in [0,1]^n \mid \textstyle\sum_{i=1}^n \pi_i = k\}, \tag{8}$$

where $k \in (0, n)$. Like the simplex, this set is the intersection of the unit cube and a hyperplane. The location of the hyperplane is determined by $k$, as shown in three dimensions in Figure 2. We argue that this should be the codomain of our relaxed top-$k$ operator. As we will see in §2.2, it is an established representation of $k$-hot probabilities.

**The sigmoid top-$k$ function.** We propose the **sigmoid top-$k$** function as a differentiable relaxation of the top-$k$ operator. For $\boldsymbol{x} \in \mathbb{R}^n$ and $k \in (0, n)$ we define $\sigma_k : \mathbb{R}^n \to \Delta_k^{n-1}$ as

$$\sigma_k(\boldsymbol{x}) := \sigma(\boldsymbol{x} + c\mathbf{1}), \text{ where } c \in \mathbb{R} \text{ solves } \textstyle\sum_{i=1}^n \sigma(x_i + c) = k. \tag{9}$$

We prove the existence and uniqueness of $c$ in Appendix A. The root is found numerically by scalar root-finding Kong et al. (2020). The time complexity is $\mathcal{O}(n)$ per root-finding iteration, of which only a handful are typically required. Empirically, we observe that the number of root-finding iterations required does not grow with $n$. This is explained intuitively by $c$ being a scalar regardless of $n$. The number of iterations is also independent of $k$, which simply shifts the location of the root. This means $\sigma_k$ can be computed efficiently in high dimensions, see Figure 6. The sum-constraint implicitly defines $c$ as of $\boldsymbol{x}$ and $k$, i.e., $c(\boldsymbol{x}, k)$. Using the chain rule and implicit differentiation, we get:

$$\frac{\mathrm{d}\sigma_k(\boldsymbol{x})}{\mathrm{d}\boldsymbol{x}} = \frac{\partial\sigma_k(\boldsymbol{x})}{\partial\boldsymbol{x}} + \frac{\partial\sigma_k(\boldsymbol{x})}{\partial c}\frac{\partial c}{\partial\boldsymbol{x}} = \text{diag}(\sigma'(\boldsymbol{x} + c)) - \frac{\sigma'(\boldsymbol{x} + c)\sigma'(\boldsymbol{x} + c)^\top}{\sum_{i=1}^n \sigma'(x_i + c)} \tag{10}$$

$$\frac{\mathrm{d}\sigma_k(\boldsymbol{x})}{\mathrm{d}k} = \frac{\partial\sigma_k(\boldsymbol{x})}{\partial c}\frac{\partial c}{\partial k} = \frac{\sigma'(\boldsymbol{x} + c)}{\sum_{i=1}^n \sigma'(x_i + c)} \tag{11}$$

where $\sigma'(x) = \sigma(x)(1 - \sigma(x))$. See Appendix A for the derivation. By using implicit differentiation to find derivatives of $c$, we avoid unrolling the steps of the root-finding algorithm and differentiating

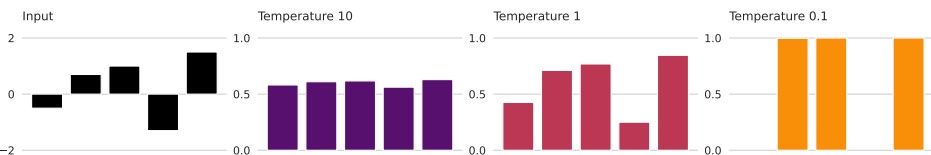

Figure 3: **Temperature scaling.** Just like the standard sigmoid and softmax functions, sigmoid top-$k$ can be tempered by dividing its input by a temperature $\tau \in \mathbb{R}_+$. Its output approaches uniform ($\pi_i = \frac{k}{n}$) as $\tau \to \infty$, and its output approaches top-$k$ as $\tau \to 0$. In this example, $n = 5$ and $k = 3$.

through them. This uses less memory and computation than unrolling while often producing a more accurate solution (Blondel et al., 2022). Note that sigmoid top-$k$ is also differentiable with respect to $k$, which can optionally be treated as learnable.

**Proposition 1.** Let $\boldsymbol{x} \in \mathbb{R}^n$, $k \in (0, n)$, and $\tau \in \mathbb{R}_+$, then the following properties hold:

    a) Order-preservation. $\sigma_k(\boldsymbol{x})_i > \sigma_k(\boldsymbol{x})_j$ if and only if $x_i > x_j$, for all $i \neq j$.

    b) Shift-invariance. $\sigma_k(\boldsymbol{x} + \alpha) = \sigma_k(\boldsymbol{x})$, for all $\alpha \in \mathbb{R}$.

    c) Invertible up to an additive constant. $\sigma^{-1}(\sigma_k(\boldsymbol{x})) = \boldsymbol{x} + c$.

    d) Infinite temperature limit. $\lim_{\tau \to \infty} \sigma_k \left(\frac{x}{\tau}\right)_i = \frac{k}{n}$.

    e) Zero temperature limit. $\lim_{\tau \to 0} \sigma_k \left(\frac{\boldsymbol{x}}{\tau}\right) = \text{top-}k(\boldsymbol{x})$ for distinct $x_i$ and $k \in \mathbb{N}$.

*Proof.* The proposition follows by definition, see Appendix A. $\qquad\square$

We list some important properties of sigmoid top-$k$ in Proposition 1. All of the properties are shared with softmax, or have softmax counterparts. Figure 3 shows a concrete example of tempering sigmoid top-$k$. The temperature plays an important role in relaxed sampling, controlling the bias–variance trade-off (Maddison et al., 2017; Jang et al., 2017). It can be fixed, annealed, or learned.

**Proposition 2.** Let $H(p) = -p \log p - (1-p) \log(1-p)$ be the binary entropy function and $\boldsymbol{x} \in \mathbb{R}^n$. Sigmoid top-$k$ solves the optimization problem:

$$\sigma_k(\boldsymbol{x}) = \arg\max_{\boldsymbol{\pi} \in \Delta_k^{n-1}} \boldsymbol{x}^\top \boldsymbol{\pi} + \sum_{i=1}^n H(\pi_i)$$

*Proof.* The proposition is derived from the Lagrangian, see Appendix A. $\qquad\square$

**Corollary 1.** In temperature-scaled sigmoid top-$k$, the temperature $\tau \in \mathbb{R}_+$ directly controls the entropy regularization strength in the optimization problem in Proposition 2

*Proof.* The corollary follows from Proposition 2, see Appendix A. $\qquad\square$

The optimization problem in Proposition 2 is an instance of a regularized prediction function (Blondel et al., 2019). Intuitively, the optimization problem maximizes the similarity between the input $\boldsymbol{x}$ and output probabilities $\boldsymbol{\pi}$ as measured by the inner product. The projection is regularized by the elementwise binary entropy, which is maximized at the center of $\Delta_k^{n-1}$ for any $k$. Figure 4 shows a comparison of differentiable relaxations of top-$k$ for the special case when $k = 1$. This way, they can be visualized alongside softmax.

## 2.2 Sampling $k$-hot with $\pi$PS sampling

**Sampling designs.** One-hot sampling draws samples $\boldsymbol{x} \in \{0, 1\}_1^n$ with parameters $\boldsymbol{\pi} \in \Delta^{n-1}$. These parameters define the categorical distribution by its **inclusion probabilities**,

$$p(x_i) = \sum_{\boldsymbol{x} \in \{0,1\}_1^n} p(\boldsymbol{x}) \mathbf{1}_{x_i = 1}. \tag{12}$$

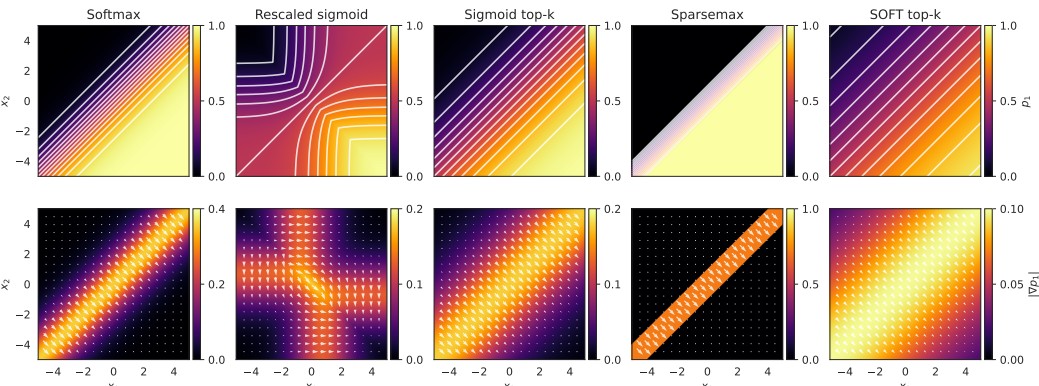

Figure 4: **Differentiable projections onto the 1D simplex.** The value (top row) and gradient (bottom row) of differentiable projections onto the simplex. In this special case of the $k$-capped simplex, when $k = 1$, we can compare against softmax. However, softmax is not applicable to the general problem. Although the functions are vector-valued, mapping $x_1$ and $x_2$ to $\pi_1$ and $\pi_2$, we only need to consider $\pi_1$, since $\pi_2 = 1 - \pi_1$.

In other words, $p(x_i)$ is the marginal probability that $x_i$ equals one. In turn, $\boldsymbol{\pi}$ is often differentiably parameterized as $\boldsymbol{\pi} = \mathrm{softmax}(\boldsymbol{\theta})$, where $\boldsymbol{\theta} \in \mathbb{R}^n$ are learnable parameters. We want to generalize this to the $k$-hot case. A natural choice is $\pi$ps sampling (Tillé, 2006), which draws samples $\boldsymbol{x} \in \{0,1\}_k^n$ with parameters $\boldsymbol{\pi} \in \Delta_k^{n-1}$. In $\pi$ps sampling, a projection onto $\Delta_k^{n-1}$ is commonly computed with a non-differentiable algorithm[1]. We propose using a top-$k$ relaxation, such as sigmoid top-$k$, to parameterize the sampling parameters as $\boldsymbol{\pi} = \sigma_k(\boldsymbol{\theta})$.

In the one-hot case, the categorical distribution is uniquely defined by its inclusion probabilities. This is not the case for the $k$-hot counterpart of $\pi$ps sampling. The categorical distribution's sample space contains $n$ elements, and its parameters are $n$-dimensional. On the other hand, $\pi$ps sampling defines a distribution over a sample space with $\binom{n}{k}$ elements, still with only $n$-dimensional parameters. This means there is not one unique distribution, but a plethora of different options known as $\pi$ps **sampling designs**. Furthermore, the same sampling design can often be implemented using different algorithms, or **sampling procedures**, which we will discuss in the next section. First, let us review some sampling designs.

Perhaps the most common design used in machine learning is weighted random sampling (Yates & Grundy, 1953), which is often implemented as Gumbel top-$k$ sampling (Kool et al., 2019). Both the papers' original authors and Tillé (2023) point out that its actual inclusion probabilities do not equal its parameters, i.e.,

$$p(x_i) = \pi_i, \tag{13}$$

does *not* hold. Worse yet, the actual inclusion probabilities are intractable, which limits both interpretability and modeling (you cannot, e.g., compute a KL-divergence with unknown probabilities). Another design used in machine learning is conditional Poisson sampling. It is the independent Bernoulli distribution[2] conditioned on $\sum_{i=1}^{n} x_i = k$. This design does not produce exact inclusion probabilities either. However, the actual inclusion probabilities can be computed. New parameters that produce the desired inclusion probabilities can be computed via numerical optimization or approximated analytically to correct the design (Chen et al., 1994; Aires, 1999; Tillé, 2006; Lundquist, 2009). There are many other sampling designs that have seen little to no use in machine learning thus far, some of which produce exact inclusion probabilities (Sampford, 1967; Brewer, 1975; Rosén, 1997). The choice of sampling design may be situation-dependent. We expand on this choice in Appendix B.

**Sampling procedures.** A sampling procedure is an algorithm that implements a sampling design. As mentioned previously, the same design may be implemented by multiple sampling procedures. In a machine learning setting, we often require rapid sampling, making efficient sampling procedures vital. We summarize some common types of procedures. **Draw-by-draw** procedures add one

---

[1]Like the one implemented in the R package `sampling` (Tillé & Matei, 2023).

[2]Independent Bernoulli sampling is known as Poisson sampling in the sampling design literature.

element to the sample at a time. They are often easy to implement, but have a time-complexity of $\mathcal{O}(nk)$ due to drawing $k$ samples sequentially. **Rejection sampling** procedures make repeated attempts to accept samples based on certain criteria. For example, conditional Poisson sampling can be implemented by drawing independent Bernoulli samples until the sample's sum is $k$. The time complexity of these algorithms depends on the expected number of iterations until acceptance, which in turn depends on the parameter values. In a learning setting, these values change. We note that there may be a substantial risk of encountering cases with low acceptance rates as a result. **Order sampling** procedures sample ranking variables and pick the top-$k$ of these. Assuming the ranking variables can be computed efficiently, the algorithm is as fast as top-$k$. Gumbel top-$k$ is an example of such an algorithm. Not that there are many other procedures that do not fall into the categories above. We elaborate on the choice of sampling procedure in Appendix B.

### 2.3 SAMPLING WITH IMPLICIT $k$

**Implicit parametrization.** Sigmoid top-$k$ is useful when the subset size is explicit. However, in some situations, we might want to learn the subset size alongside the inclusion probabilities. In this case, we can simply parameterize the desired inclusion probabilities with a regular sigmoid function and define $k$ implicitly. It is always possible to let $k = \sum_{i=1}^{n} \sigma(\theta_i)$, since any point in $(0,1)^n$ lies in $\Delta_k^{n-1}$ for some $k$. However, this produces non-integer $k$ which are not compatible with standard $\pi$ps sampling methods. If the subset size is learned, we may want to regularize it. This can be done by simply adding a loss term like $\lambda \sum_{i=1}^{n} \pi_i$, where $\lambda$ controls the subset size, or sparsity level, similar to the regularization parameter in lasso regression (Tibshirani, 1996).

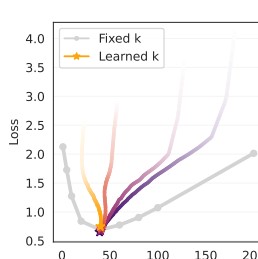

Figure 5: **Learning $k$.** Feature selection for MNIST classification with the regularization term $\lambda \sum_{i=1}^{n} \pi_i$ and $\lambda = 0.01$. Learning $k$ finds a near-optimal $k$ in a single run for different initial guesses.

**Phantom unit.** To sample with non-integer $k$ without introducing rounding errors, we propose using a **phantom unit** (Bondesson & Grafström, 2011). A phantom unit is a temporary variable with inclusion probability,

$$\pi_{n+1} := 1 - \left(\sum_{i=1}^{n} \pi_i - \left\lfloor \sum_{i=1}^{n} \pi_i \right\rfloor\right), \qquad (14)$$

that is concatenated to the existing variables. These extended inclusion probabilities sum to the integer $1 + \lfloor \sum_{i=1}^{n} \pi_i \rfloor$. This way, $\pi$ps sampling can be performed as usual using the extended inclusion probabilities, after which the element corresponding to the phantom unit is removed from the sample. This results in two possible subset sizes with probabilities given by:

$$\Pr\left(\sum_{i=1}^{n} x_i = k\right) = \begin{cases} \pi_{n+1}, & k = 1 + \lfloor \sum_{i=1}^{n} \pi_i \rfloor \\ 1 - \pi_{n+1}, & k = \lfloor \sum_{i=1}^{n} \pi_i \rfloor \end{cases} \qquad (15)$$

Note that the original inclusion probabilities were not modified, guaranteeing exact inclusion probabilities. Algorithm 2 details how to implement differentiable sampling with implicit $k$ in pseudocode. Sampling with a phantom unit is also useful if sigmoid top-$k$ is used with a non-integer $k$.

### 2.4 GRADIENT ESTIMATION

**Straight-through.** Straight-through estimation (Bengio et al., 2013) is a simple and popular approach to gradient estimation. As the name suggests, the sample's gradient is copied "straight-through" to its input. Intuitively, in the one-hot case, this replaces the categorical sample $\boldsymbol{x}$ by its expected value $\mathbb{E}[\boldsymbol{x}] = \boldsymbol{\pi}$ in the backward pass. This was recently shown to be a first-order approximation for categorical samples (Liu et al., 2023). In the $k$-hot case, we can make the same replacement. For $\pi$ps sampling with exact inclusion probabilities, we have $\mathbb{E}[\boldsymbol{x}] = \boldsymbol{\pi}$, just like for the categorical distribution.

**Proposition 3.** The expected gradient of the straight-through estimator is a first-order approximation of the true gradient for $k$-hot sampling.

*Proof.* The proposition is shown by extending the proof in Liu et al. (2023), see Appendix C. □

---

**Algorithm 1** Sample $k$-hot with explicit $k$

---

**Require:** $\boldsymbol{\theta} \in \mathbb{R}^n, k \in \mathbb{N}, 1 < k < n$
1: $\boldsymbol{\pi} \leftarrow \sigma_k(\boldsymbol{\theta})$
2: Sample $\boldsymbol{x} \in \{0,1\}_k^n$ such that $p(x_i) = \pi_i$           $\triangleright$ By $\pi$ps sampling
3: **return** $\boldsymbol{x}$

---

**Algorithm 2** Sample $k$-hot with implicit $k$

---

**Require:** $\boldsymbol{\theta} \in \mathbb{R}^n$
1: $\boldsymbol{\pi} \leftarrow \sigma(\boldsymbol{\theta})$
2: $k \leftarrow \sum_{i=1}^n \pi_i$
3: Sample $\boldsymbol{x} \in \{0,1\}_{\lfloor k \rfloor}^n \cup \{0,1\}_{\lfloor k \rfloor + 1}^n$ such that $p(x_i) = \pi_i$    $\triangleright$ By $\pi$ps sampling w. phantom unit
4: **return** $\boldsymbol{x}$

---

**Algorithm 3** Sample relaxed $k$-hot

---

**Require:** $\boldsymbol{\theta} \in \mathbb{R}^n, k \in \mathbb{N}, 1 < k < n$
1: Sample $g_i \sim \text{Gumbel}(0,1)$ for $i = 1, \ldots, n$
2: $\boldsymbol{x} \leftarrow \sigma_k(\boldsymbol{\theta} + \boldsymbol{g})$
3: **return** $\boldsymbol{x}$

---

Algorithm 1 and Algorithm 2 use straight-through estimation. We also extend ReinMax (Liu et al., 2023) to $k$-hot sampling in Appendix C.

**Score function estimators.** For score function estimators, we need to use a $\pi$ps sampling design with a tractable score function. Clearly, this is possible if we can compute $p(\boldsymbol{x})$ and differentiate. For instance, Wijk et al. (2025) proposed a score function estimator for conditional Poisson sampling where the conditional term is computed using a discrete Fourier transform.

### 2.5 RELAXED $k$-HOT SAMPLING

Another approach to sampling is solving optimization problems with random perturbations drawn from the Gumbel distribution (Gumbel, 1954; Huijben et al., 2023). For instance, a categorical sample can be drawn from logits $\boldsymbol{\theta} \in \mathbb{R}^n$ as

$$\boldsymbol{x} = \text{argmax}(\boldsymbol{\theta} + \boldsymbol{g}), \quad \boldsymbol{g} \sim \text{Gumbel}(0,1). \tag{16}$$

Then, replacing argmax by its differentiable relaxation, softmax, we get the Gumbel–softmax (Maddison et al., 2017; Jang et al., 2017) distribution over $\Delta^{n-1}$. These relaxed samples can be differentiated using the reparametrization trick (Kingma & Welling, 2014). The resulting gradients can also be used as estimates for the hard sample's gradient. This can be generalized to the $k$-hot case by substituting argmax and softmax with top-$k$ and a top-$k$ relaxation (Kool et al., 2020; Xie & Ermon, 2019). Here, the top-$k$ relaxation used is critical. We find that using sigmoid top-$k$ improves this approach significantly, reducing its time complexity from $\mathcal{O}(nk)$ to $\mathcal{O}(n)$ and producing better experimental results compared to the original work. The pseudocode for relaxed sampling is given in Algorithm 3.

## 3 RELATED WORK

**Relaxed top-$k$.** The top-$k$ relaxation used in Xie & Ermon (2019); Plötz & Roth (2018) works by sequentially applying softmax and masking the input. The relaxation can be tempered, but the authors point out that ordering is not preserved at all temperatures (unlike sigmoid top-$k$, see Proposition 1). The relaxation's output does not lie in $\Delta_k^{n-1}$, as elements may exceed one[3]. Moreover, its time complexity is $\mathcal{O}(nk)$ due to the $k$ sequential steps. Pervez et al. (2023) rescale the sigmoid's output linearly to enforce the sum-constraint. This closed-form solution is fast and easy to

---

[3]Algorithm 2 in Xie & Ermon (2019) incorrectly states that the output lies in $[0,1]^n$, which is disproven by a counterexample on the following page where an output is $[1.05, 0.95]$.

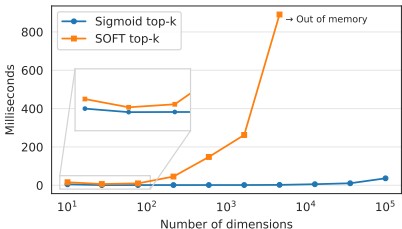 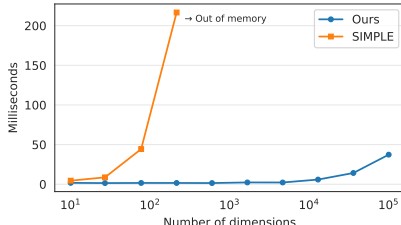

Figure 6: **Runtime. Left:** Relaxed top-$k$ wall-clock time comparison between sigmoid top-$k$ and SOFT top-$k$. The experimental setup is the same as for the root-finding benchmark in Appendix A. **Right:** Differentiable $k$-hot sampling wall-clock time comparison between Algorithm 1 and SIMPLE. SIMPLE is using warmed-up `torch.compile` and loading precomputed values from disk. Both comparisons use the official implementations and measure both forward and backward times.

implement, but lacks shift-invariance (unlike sigmoid top-$k$, see Proposition 1 and Figure 4) and is non-differentiable along the boundary of its piecewise definition.

Multiple previous works have proposed relaxed top-$k$ functions as solutions to optimization problems. A common theme is to introduce a regularization term. Without regularization, the solutions would be sparse, since many points would map to vertices of the convex polytope $\Delta_k^{n-1}$. In fact, optimizing $\|\boldsymbol{x} - \boldsymbol{\pi}\|^2$ over $\Delta^{n-1}$ is sparsemax (Martins & Astudillo, 2016). SOFT top-$k$ (Xie et al., 2020) solves an entropy-regularized optimal transport problem using the Sinkhorn algorithm. Similarly, Sander et al. (2023) propose a top-$k$ approach capable of producing sparse solutions, which is solved using, e.g., Dykstra's algorithm. Both of the aforementioned algorithms use implicit differentiation of the KKT conditions to derive the Jacobian. Proposition 2 shows that sigmoid top-$k$ also solves an optimization problem. Compared to the above, the solution is found by a simpler algorithm: scalar root-finding. See Figure 6 for a wall-clock comparison between SOFT top-$k$ and sigmoid top-$k$.

**Shifted sigmoids.** Liu et al. (2024) propose a differentiable top-$k$ approach that is closely related to sigmoid top-$k$ . They propose an input-shifted sigmoid where the shift is learned, and set the temperature to a low value. In sigmoid top-$k$, the sum constraint is used to control the subset size directly, so that the shift is input-dependent and guarantees the same integer $k$ for all inputs. On the topic of shifted sigmoids, Ramapuram et al. (2025) used a constant shift of $-\log(n)$ in sigmoid attention to approximately normalize its output, i.e., project onto $\Delta^{n-1}$ like softmax.

**Differentiable $k$-hot sampling.** As discussed in §2.5, the reparameterized sampling proposed by Xie & Ermon (2019) builds upon Gumbel top-$k$ sampling (Kool et al., 2019) by replacing the top-$k$ operation with relaxed top-$k$ and using the reparametrization trick to differentiate through the Gumbel noise (Jang et al., 2017; Maddison et al., 2017). Unfortunately, the top-$k$ relaxation used (discussed in the earlier segment on relaxed top-$k$) is far from ideal. Subsequent works have used this flawed relaxation as a benchmark and may underestimate relaxed top-$k$ sampling as a result.

NCPSS (Pervez et al., 2023) is similar to our proposed differentiable $\pi$ps sampling approach. It is a rescaled sigmoid (discussed in the earlier segment on relaxed top-$k$) as its differentiable relaxation of top-$k$. The proposed sampling procedure, iterative Poisson sampling, does not sample exactly $k$-hot vectors. The procedure is reminiscent of repeated Poisson sampling (Grafström, 2009), which iterates until the sample is exactly $k$-hot instead. SIMPLE (Ahmed et al., 2023) computes the marginal probabilities per element using dynamic programming and uses these to estimate the sample's gradient. Viewed as $\pi$ps sampling, the $k$-hot distribution in SIMPLE is conditional Poisson sampling, and the marginal probabilities are the inclusion probabilities. Recursive methods to compute said inclusion probabilities have been proposed previously Chen et al. (1994). Moreover, using the inclusion probabilities as gradient estimates is equivalent to straight-through estimation if sampling is done with exact inclusion probabilities. We find that computing the inclusion probabilities using dynamic programming is less efficient than sampling according to Algorithm 1 with straight-through estimation, see Figure 6.

Table 2: **Top-$k$ relaxations.** An overview of recent top-$k$ relaxations, their forward- and backward-passes.

| Method | Foward | Backward | Comment |
|---|---|---|---|
| Renorm. sigmoid (Pervez et al., 2023) | Closed form | Standard | ✗ Not shift-invariant |
| Sequential softmax (Xie & Ermon, 2019) | Softmax $k$ times | Unrolled | ✗ Outputs may exceed 1 |
| SOFT top-$k$ (Xie et al., 2020) | Optimal transport | Implicit | |
| Sparse soft top-$k$ (Sander et al., 2023) | Isotonic optmization | Implicit | |
| Sigmoid top-$k$ (Ours) | Scalar root-finding | Implicit | ✓ Simple and scalable |

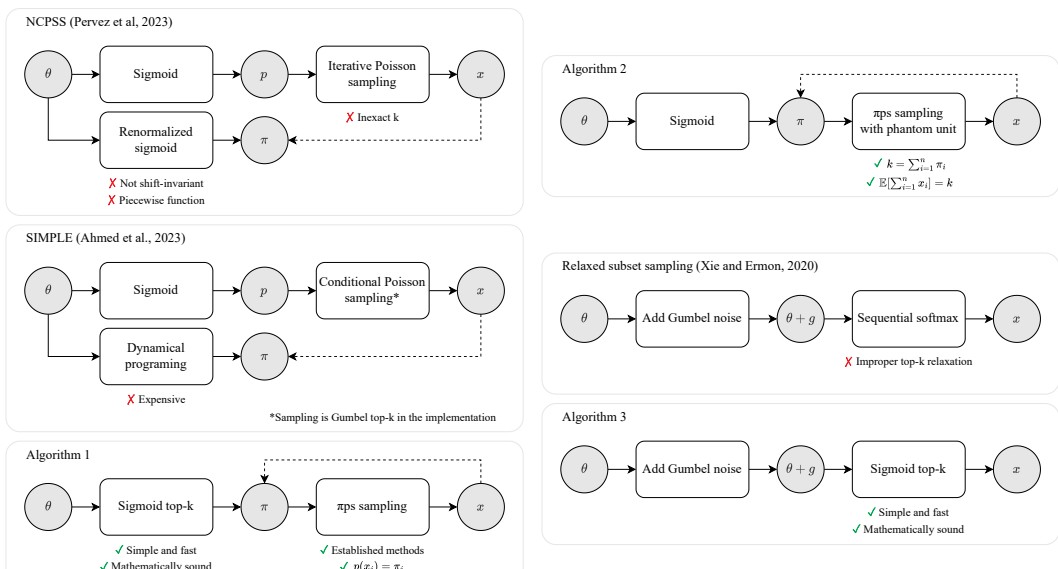

Figure 7: **Differentiable top-$k$ sampling.** A graphical overview comparing previous works and our proposed algorithms. NCPSS and SIMPLE use Bernoulli probabilities $\boldsymbol{p} \in [0,1]^n$ in the forward pass, and compute . Our approach, Algorithm 1, parameterizes the desired inclusion probabilities in the forward pass, leading to a streamlined approach that mirrors one-hot sampling (see Figure 1).

SFESS (Wijk et al., 2025), like SIMPLE, considers conditional Poisson sampling. The score function is calculated using a discrete Fourier transform (Fernandez & Williams, 2010) and is used in a score function estimator. We note that the product of inclusion probabilities could be used to compute the score function instead.

## 4 EXPERIMENTS

**Overview.** We validate our approach on two tasks: feature selection and sparse representation learning. In the results tables $\pi$ST refers to $\pi$ps sampling with a straight-through gradient estimate (Algorithm 1) and $\pi$ST2 to the version in Appendix C. Gumbel–$\sigma_k$ refers to relaxed samplng with sigmoid top-$k$ with hard samples and relaxed gradients (Algorithm 3). We report results for the MNIST and Fashion-MNIST datasets (LeCun et al., 1998; Xiao et al., 2017). See Appendix D for details on network architectures and hyperparameters.

**Feature selection.** Differentiable $k$-hot sampling can be used to jointly optimize an arbitrary neural network and a distribution of $k$-hot input masks (Huijben et al., 2019). Results are shown in Table 3. To demonstrate learning with an implicit subset size (Algorithm 2), we consider feature selection with a penalized loss. The results (Figure 5) show how this approach can optimize the penalized loss in a single run.

**Variational autoencoders.** Variational autoencoders (Kingma & Welling, 2014) with a $k$-hot latent space. Because not every method has tractable inclusion probabilities, we follow previous

Table 3: **Feature selection.** Test loss for feature selection with $n = 784$ features and $k = 50$ selections. Results are shown with one standard deviation computed from five different random seeds.

| Method | MNIST | Fashion-MNIST |
|---|---|---|
| Xie & Ermon (2019) | $0.113 \pm 2.44$e-03 | $0.300 \pm 2.39$e-03 |
| NCPSS | $0.134 \pm 2.47$e-03 | $0.317 \pm 8.13$e-03 |
| SIMPLE | $0.099 \pm 9.03$e-04 | $0.287 \pm 4.08$e-04 |
| $\pi$ST | $0.102 \pm 1.35$e-03 | $0.291 \pm 1.20$e-03 |
| $\pi$ST2 | $0.098 \pm 5.49$e-04 | $\mathbf{0.285 \pm 2.86}$**e-04** |
| Gumbel–$\sigma_k$ | $\mathbf{0.096 \pm 5.06}$**e-04** | $0.286 \pm 4.89$e-04 |

Table 4: **Variational autoencoders.** Test loss for a small VAE with a latent space of ten $k$-hot vectors ($n = 10$ and $k = 5$) and a large VAE with a single $k$-hot vector ($n = 1000$ and $k = 500$). Results are shown with one standard deviation computed from five different random seeds.

| Method | $n = 10$ and $k = 5$ ($\times 10$) | | $n = 1000$ and $k = 500$ ($\times 1$) | |
|---|---|---|---|---|
| | MNIST | Fashion-MNIST | MNIST | Fashion-MNIST |
| Xie & Ermon (2019) | $98.26 \pm 2.63$e-00 | $234.94 \pm 4.42$e-01 | – | – |
| NCPSS | $82.88 \pm 1.87$e-01 | $226.37 \pm 3.27$e-01 | $66.25 \pm 2.12$e-01 | $223.94 \pm 1.71$e-00 |
| SIMPLE | $\mathbf{81.73 \pm 2.07}$**e-01** | $\mathbf{225.16 \pm 1.22}$**e-01** | – | – |
| $\pi$ST | $82.78 \pm 2.30$e-01 | $226.29 \pm 2.49$e-01 | $66.04 \pm 2.06$e-01 | $222.38 \pm 7.96$e-01 |
| $\pi$ST2 | $86.83 \pm 6.65$e-01 | $228.97 \pm 1.91$e-01 | $\underline{65.49 \pm 1.59}$e-01 | $\mathbf{217.19 \pm 2.60}$**e-01** |
| Gumbel–$\sigma_k$ | $\underline{82.06 \pm 2.12}$e-01 | $225.80 \pm 2.06$e-01 | $\mathbf{63.46 \pm 1.73}$**e-01** | $217.59 \pm 2.74$e-01 |

works and compute the prior term as the KL-divergence between $\mathrm{softmax}(\boldsymbol{\theta})$ and uniform (Wijk et al., 2025). Previous works used latent spaces with multiple small $k$-hot distributions ($n = 10$). To demonstrate our scalable approach, we also consider a latent space with a single large $k$-hot distribution ($n = 1000$). Results are shown in Table 4. For the larger $n$, SIMPLE runs out of memory and the sequential softmax used in Xie & Ermon (2019) faces a significant slowdown due to its $\mathcal{O}(nk)$ time-complexity.

**Additional experiments.** Experiments evaluating different root-finding algorithms for sigmoid top-$k$ are presented in Appendix B. We also compare the wall-clock time of SOFT top-$k$ (Xie et al., 2020) and sigmoid top-$k$ as well as Algorithm 1 and SIMPLE (Ahmed et al., 2023) in Figure 6.

## 5 CONCLUSION

In this work, we proposed a framework for differentiable top-$k$ by generalizing from one-hot to $k$-hot. By broadening the perspective to both relaxations and sampling, we identify top-$k$ relaxations and $\pi$ps sampling as key components of multiple algorithms. These components, along with a principled straight-through estimator, pave the way for future work on improved estimators. At the same time, we cast previous works on differentiable $k$-hot sampling into this framework, offering a modular understanding that reveals their limitations. Finally, we proposed sigmoid top-$k$, an efficient and fully differentiable generalization of softmax that scales to larger problems. In total, our work establishes a foundation for more powerful and scalable differentiable top-$k$.

## REPRODUCIBILITY STATEMENT

For theoretical results, Appendix A includes proofs of Propositions 1 and 2 and a derivation of the sigmoid top-$k$ function's derivatives. Appendix C includes a proof of Proposition 3. For experimental results, Appendix D describes hardware, network architectures, and hyperparameters used in the main experiments of §4. Appendix A includes a section describing experiments on root-finding, the setup of which was also used for the comparisons in Figure 6. Code will be made available here.

## LLM USAGE

LLMs were used to give feedback on writing, including rephrasing sentences, and code for figures and equations. For coding, LLMs were used to generate starting points, autocompletion, and code for plots.

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

# A Sigmoid top-$k$

**Proof of existence and uniqueness of $c$.**

*Proof.* The existence of $c \in \mathbb{R}$ such that

$$\sum_{i=1}^{n} \sigma(x_i + c) = k, \tag{17}$$

where $\boldsymbol{x} \in \mathbb{R}^n$ and $k \in (0, n)$ is easily seen using the intermediate value theorem. $f(c) = \sum_{i=1}^{n} \sigma(\boldsymbol{x} + c)$ is continuous since it is a sum of continuous functions. As $c \to -\infty$, $f(c) \to 0$, and as $c \to \infty$, $f(c) \to n$. By the intermediate value theorem, there exists $c$ such that $f(c) = k$, since $0 < k < n$. The uniqueness of this solution follows from the fact that $f(c)$ is strictly increasing, which in turn follows from it being a sum of strictly increasing functions. $\square$

**Derivation of derivatives.** First, we use implicit differentiation to derive derivatives of $c$ from the sum-constraint.

$$\sum_{i=1}^{n} \sigma(x_i + c) = 0 \tag{18}$$

We differentiate the sum-constraint with respect to $x_j$:

$$\frac{\partial}{\partial x_j} \sum_{i=1}^{n} \sigma(x_i + c(x, k)) = 0 \tag{19}$$

$$\sum_{i=1}^{n} \sigma'(x_i + c) \left( \frac{\partial x_i}{\partial x_j} + \frac{\partial c}{\partial x_j} \right) = 0. \tag{20}$$

Here, $\frac{\partial x_i}{\partial x_j} = 1$ for $i = j$ and $0$ otherwise. The equation simplifies as

$$\sigma'(x_j + c) + \left( \sum_{i=1}^{n} \sigma'(x_i + c) \right) \frac{\partial c}{\partial x_j} = 0 \tag{21}$$

$$\frac{\partial c}{\partial x_j} = -\frac{\sigma'(x_j + c)}{\sum_{i=1}^{n} \sigma'(x_i + c)} \tag{22}$$

We differentiate the sum-constraint with respect to $k$:

$$\frac{\partial}{\partial k} \sum_{i=1}^{n} \sigma(x_i + c(x, k)) = 1 \tag{23}$$

$$\frac{\partial c}{\partial k} \sum_{i=1}^{n} \sigma'(x_i + c) = 1 \tag{24}$$

$$\frac{\partial c}{\partial k} = \frac{1}{\sum_{i=1}^{n} \sigma'(x_i + c)}. \tag{25}$$

Now, we move on to differentiating the function

$$\sigma_k(\boldsymbol{x}) = \sigma(\boldsymbol{x} + c\mathbf{1}). \tag{26}$$

We differentiate with respect to $x_i$. Using the chain rule:

$$\frac{\mathrm{d}\sigma_k(\boldsymbol{x})_i}{\mathrm{d}x_j} = \sigma'(x_i + c) \left( \frac{\partial x_i}{\partial x_j} + \frac{\partial c}{\partial x_j} \right). \tag{27}$$

Again, $\frac{\partial x_i}{\partial x_j} = 1$ for $i = j$ and $0$ otherwise. We substitute our previously derived $\frac{\partial c}{\partial x_j}$,

$$\frac{\mathrm{d}\sigma_k(\boldsymbol{x})_i}{\mathrm{d}x_j} = \begin{cases} \sigma'(x_i + c) - \sigma'(x_i + c) \dfrac{\sigma'(x_i + c)}{\sum_{l=1}^{n} \sigma'(x_l + c)} & \text{if } i = j, \\ -\sigma'(x_i + c) \dfrac{\sigma'(x_j + c)}{\sum_{l=1}^{n} \sigma'(x_l + c)} & \text{if } i \neq j. \end{cases} \tag{28}$$

Or, using vector notation,

$$\frac{\mathrm{d}\sigma_k(\boldsymbol{x})}{\mathrm{d}x} = \mathrm{diag}(\sigma'(\boldsymbol{x} + c)) - \frac{\sigma'(\boldsymbol{x} + c)\,\sigma'(\boldsymbol{x} + c)^\top}{\sum_{i=1}^{n} \sigma'(x_i + c)} \tag{29}$$

Finally, we differentiate with respect to $k$ and substitute $\frac{\partial c}{\partial k}$,

$$\frac{\mathrm{d}\sigma_k(\boldsymbol{x})}{\mathrm{d}k} = \sigma'(\boldsymbol{x} + c)\frac{\partial c}{\partial k} \tag{30}$$

$$\frac{\mathrm{d}\sigma_k(\boldsymbol{x})}{\mathrm{d}k} = \frac{\sigma'(\boldsymbol{x} + c)}{\sum_{i=1}^{n}\sigma'(x_i + c)} \tag{31}$$

**Proof of Proposition 1.**

*Proof.* Properties a) to c) are easily seen:

    a) Follows directly from $\sigma(x_i + c)$ being strictly increasing.

    b) $\sigma_k(\boldsymbol{x} + \alpha) = \sigma(\boldsymbol{x} + \alpha + c) = \sigma_k(\boldsymbol{x})$.

    c) $\sigma^{-1}(\sigma_k(\boldsymbol{x})) = \sigma^{-1}(\sigma(\boldsymbol{x} + c)) = \boldsymbol{x} + c$.

    d) $\lim_{\tau\to\infty}\frac{x_i}{\tau} = 0$ so the sum-constraint requires $\lim_{\tau\to\infty}\frac{x_i+c}{\tau} = \sigma^{-1}(k/n)$.

Finally, for e) we know that

$$\lim_{\tau\to 0}\sigma\left(\frac{x}{\tau}\right) = \begin{cases} 1, & x > 0 \\ 0, & x < 0 \end{cases} \tag{32}$$

Next, the sum-constraint $\sum_{i=1}^{n}\sigma\left(\frac{x_i+c}{\tau}\right) = k$, so $\lim_{\tau\to 0}\sigma\left(\frac{x_i+c}{\tau}\right) = 1$ for exactly $k$ indices. Consider $x_i$ in sorted order

$$x_1 > \cdots > x_k > x_{k+1} > \cdots > x_n,$$

Because the sigmoid function is increasing, the order is preserved for its outputs. It follows that all $\lim_{\tau\to 0}\sigma\left(\frac{x_i+c}{\tau}\right) = 1$ for $i \geq k$ and 0 otherwise.

$$\underbrace{\sigma\left(\frac{x_1 + c}{\tau}\right) > \cdots > \sigma\left(\frac{x_k + c}{\tau}\right)}_{\to 1} > \underbrace{\sigma\left(\frac{x_{k+1} + c}{\tau}\right) > \cdots > \sigma\left(\frac{x_n + c}{\tau}\right)}_{\to 0},$$

This is exactly top-$k(\boldsymbol{x})$. $\qquad\square$

**Proof of Proposition 2.**

*Proof.* We consider the optimization problem

$$\sigma_k(\boldsymbol{x}) = \arg\max_{\boldsymbol{\pi}\in\Delta_k^{n-1}} \boldsymbol{x}^T\boldsymbol{\pi} + \sum_{i=1}^{n} H(\pi_i) \tag{33}$$

$$= \arg\min_{\substack{\boldsymbol{\pi}\in(0,1)^n \\ \sum_{i=1}^{n}\pi_i=k}} \sum_{i=1}^{n}\log(1 - \pi_i) + \pi_i\log\frac{\pi_i}{1 - \pi_i} - x_i\pi_i \tag{34}$$

Here, we limited $\boldsymbol{\pi}$ to $(0,1)^n$ for simplicity. First, we note that the feasible set is convex as it is the intersection of two convex sets. For each $i$, the elementwise objective and its derivatives

$$f_i(\pi_i) = \log(1 - \pi_i) + \pi_i\log\frac{\pi_i}{1 - \pi_i} - x_i\pi_i \tag{35}$$

$$f_i'(\pi_i) = \log\frac{\pi_i}{1 - \pi_i} - x_i \tag{36}$$

$$f_i''(\pi_i) = \frac{1}{\pi_i(1 - \pi_i)} \tag{37}$$

$f_i''(\pi_i) > 0$ over the feasible set, so $f_i(\pi_i)$ is convex. The sum of convex functions is convex, so the original objective's derivative is convex. Since the feasible set and objective are convex,

the optimization problem is convex. This implies that it has a unique solution. Next, we have the Lagrangian for $\boldsymbol{\pi} \in (0,1)^n$

$$\mathcal{L}(\boldsymbol{\pi}, c) = \sum_{i=1}^{n} \left( \log(1 - \pi_i) + \pi_i \log \frac{\pi_i}{1 - \pi_i} - x_i \pi_i \right) + c \left( \sum_{i=1}^{n} \pi_i - k \right) \tag{38}$$

The elementwise stationarity condition $\frac{\partial \mathcal{L}}{\partial \pi_i} = 0$ is

$$\sigma^{-1}(\pi_i) = x_i + c \tag{39}$$

$$\pi_i = \sigma(x_i + c) \tag{40}$$

which retrieves the scalar shift in sigmoid top-$k$. Finally, at the stationarity condition, the sum constraint becomes

$$\sum_{i=1}^{n} \sigma(x_i + c) - k = 0, \tag{41}$$

exactly the equation that defines $c$ in sigmoid top-$k$. It has a unique solution, which we have already proven in the first paragraph of this appendix. $\qquad\square$

**Proof of Corollary 1.**

$$\sigma_k \left( \frac{\boldsymbol{x}}{\tau} \right) = \arg\max_{\boldsymbol{\pi} \in \Delta_k^{n-1}} \frac{\boldsymbol{x}^T \boldsymbol{\pi}}{\tau} + \sum_{i=1}^{n} H(\pi_i) = \arg\max_{\boldsymbol{\pi} \in \Delta_k^{n-1}} \boldsymbol{x}^T \boldsymbol{\pi} + \tau \sum_{i=1}^{n} H(\pi_i) \tag{42}$$

**Root-finding.** The root-finding problem can be solved using the bisection method, which guarantees linear convergence. The root is bounded by $\pm(\max_i |x_i| + \sigma^{-1}(1 - \epsilon))$ for a small $\epsilon$ that saturates the sigmoid. Finding an acceptable $c$ can be impossible with single-precision floats, especially for large $n$. We find that using double-precision just in the bisection resolves this issue at the cost of slightly increased memory use and computational overhead. The resulting $\sigma(\boldsymbol{x} + c)$ is cast back to single-precision.

Faster convergence can be achieved using a hybrid method. Since the sigmoid is an autonomous function, computing its derivatives is inexpensive. Newton's method can be much faster than bisection with quadratic convergence. However, Newton's method can diverge. We can keep the bisection method's linear worst-case performance by combining it with Newton's method. We found that simply evaluating both steps and picking the one that reduces the error the most works well. Halley's method uses the second derivative to achieve cubic convergence. We note that the first derivative is always positive, and the second is only zero if all $x_i$ are equal, in which case the root is trivial. Higher-order Householder's methods are possible, but we don't see any improvement beyond Halley's method. Finally, Newton's method benefits from a good starting guess. We pick a starting guess based on two complementary heuristics. The logit-heuristic

$$c \approx \sigma^{-1} \left( \frac{k}{n} \right) - \frac{1}{n} \sum_{i=1}^{n} x_i,$$

is accurate when the $x_i$ are approximately equal to their mean. The quantile-heurisitc

$$c \approx -x_{\left( \frac{n-k}{n} \right)},$$

where the subscript denotes the $\left( \frac{n-k}{n} \right)$-th quantile of $\boldsymbol{x}$, is accurate when the $x_i$ are spread out. Picking the one with the lowest error results in a robust starting guess. Figure 8 shows a comparison of the root-finding approaches discussed above.

## B  SAMPLING

Below are some considerations when choosing a sampling design for differentiable $\pi$ps sampling.

- Exact inclusion probabilities $p(x_i) = p_i$.

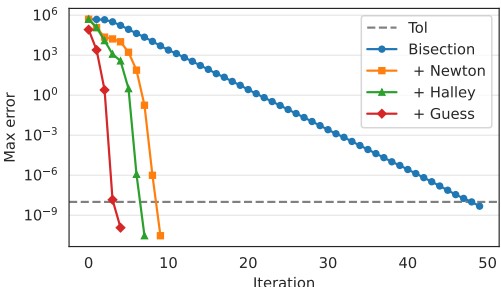

Figure 8: **Root-finding benchmark.** We solve a batch of 100 root-finding problems with $n = 10^6$ and tolerance $10^{-8}$. The instances are random with $x_i \sim \mathcal{N}(0, \sigma^2)$ where $\sigma$ is evenly spaced from 0.1 to 5 across the batches and $k \sim \mathcal{U}(1, n-1)$ for each batch. The sigmoid function $\sigma$ and standard deviation $\sigma$ should not be confused. The graph shows the maximum error across batches at each iteration.

Table 5: **Grid search results.** We evaluate different root-finding algorithms using the same experimental setup as in Figure 8, except with $n = 10^4$. The table shows the mean and standard deviation of 100 repetitions with different random batches, with the three fastest wall-clock times indicated on the right. Halley's method and choosing the best starting guess from both the logit and quantile heuristics gives both the fewest number of iterations and the fastest wall-clock time.

| Step | Guess | Iterations | Time [ms] | |
|---|---|---|---|---|
| Bisection | – | $43.0 \pm 0.00$ | $9.21 \pm 1.58$ | |
| Newton | Zero | $8.16 \pm 0.44$ | $5.62 \pm 1.19$ | |
| | Quantile | $7.27 \pm 0.73$ | $5.18 \pm 2.54$ | |
| | Logit | $7.34 \pm 0.86$ | $4.55 \pm 1.04$ | |
| | Both | $5.34 \pm 0.47$ | $3.90 \pm 0.84$ | 3rd |
| Halley | Zero | $5.82 \pm 0.41$ | $5.52 \pm 1.63$ | |
| | Quantile | $4.50 \pm 0.56$ | $4.29 \pm 0.98$ | |
| | Logit | $4.32 \pm 0.53$ | $3.66 \pm 0.87$ | 2nd |
| | Both | $3.15 \pm 0.36$ | $3.25 \pm 0.79$ | 1st |

- Tractable $p(\boldsymbol{x})$.
- High entropy.

Depending on the problem, a lower entropy design may be preferable. For example, systematic sampling can be used to design a sampling based on prior knowledge. Some sampling designs that meet all or most of these criteria are: Sampford sampling (Sampford, 1967), adjusted conditional Poisson sampling, adjusted Pareto sampling (Rosén, 1997), the designs implemented by Brewer's method (Brewer, 1975), and the pivotal method (Deville & Tillé, 1998). This is not an exhaustive list, but it should provide us with at least one good option.

The sampling procedure, i.e., the algorithm that implements the sampling design, determines how rapidly samples can be drawn. In a machine learning setting, we often draw many samples. For instance, we may draw one sample per optimization step when optimizing a model with stochastic gradient descent. Generally, we also prefer procedures that can be implemented in a vectorized fashion to run efficiently on, e.g., GPUs.

After considering these criteria, adjusted Pareto sampling (Rosén, 1997) appears to be a good option. It is a high-entropy design (Grafström, 2010) with exact inclusion probabilities (if adjusted). There is an order sampling procedure with time-complexity $\mathcal{O}(n \log k)$ that has a vectorized implementation in just a few lines of code. A caveat is that properly adjusting the inclusion probabilities can be slow. A faster option is to use a heuristic adjustment with negligible overhead.

## C  GRADIENT ESTIMATION

**Proof of Proposition 3.**

*Proof.* We expand the proof in Liu et al. (2023) to the $k$-hot case. First, we write the true gradient as a sum over all $k$-hot vectors:

$$\nabla := \frac{\partial}{\partial \boldsymbol{\theta}} \mathbb{E}[f(\boldsymbol{x})] = \frac{\mathrm{d}}{\mathrm{d}\boldsymbol{\theta}} \sum_{\boldsymbol{x} \in \{0,1\}_k^n} f(\boldsymbol{x}) p(\boldsymbol{x}) = \sum_{\boldsymbol{x} \in \{0,1\}_k^n} f(\boldsymbol{x}) \frac{\mathrm{d}p(\boldsymbol{x})}{\mathrm{d}\boldsymbol{\theta}}.$$

We can rewrite this by adding and subtracting $\mathbb{E}[f(\boldsymbol{x})]$ and rearranging the sum

$$\nabla = \sum_{\boldsymbol{x} \in \{0,1\}_k^n} (f(\boldsymbol{x}) - \mathbb{E}[f(\boldsymbol{x})] + \mathbb{E}[f(\boldsymbol{x})]) \frac{\mathrm{d}p(\boldsymbol{x})}{\mathrm{d}\boldsymbol{\theta}}$$

$$= \sum_{\boldsymbol{x} \in \{0,1\}_k^n} (f(\boldsymbol{x}) - \mathbb{E}[f(\boldsymbol{x})]) \frac{\mathrm{d}p(\boldsymbol{x})}{\mathrm{d}\boldsymbol{\theta}} + \underbrace{\sum_{\boldsymbol{x} \in \{0,1\}_k^n} \mathbb{E}[f(\boldsymbol{x})] \frac{\mathrm{d}p(\boldsymbol{x})}{\mathrm{d}\boldsymbol{\theta}}}_{=0}$$

Here, the second term is zero, since

$$\sum_{\boldsymbol{x} \in \{0,1\}_k^n} \mathbb{E}[f(\boldsymbol{x})] \frac{\mathrm{d}p(\boldsymbol{x})}{\mathrm{d}\boldsymbol{\theta}} = \mathbb{E}[f(\boldsymbol{x})] \frac{\mathrm{d}}{\mathrm{d}\boldsymbol{\theta}} \sum_{\boldsymbol{x} \in \{0,1\}_k^n} p(\boldsymbol{x}) = \mathbb{E}[f(\boldsymbol{x})] \frac{\mathrm{d}1}{\mathrm{d}\boldsymbol{\theta}} = 0.$$

We continue by expanding the expectation

$$\nabla = \sum_{\boldsymbol{x} \in \{0,1\}_k^n} \sum_{\boldsymbol{y} \in \{0,1\}_k^n} p(\boldsymbol{y})(f(\boldsymbol{x}) - f(\boldsymbol{y})) \frac{\mathrm{d}p(\boldsymbol{x})}{\mathrm{d}\boldsymbol{\theta}}$$

Next, define a first-order estimator by finite difference approximation, $f(\boldsymbol{x}) - f(\boldsymbol{y}) \approx \frac{\mathrm{d}f(\boldsymbol{y})}{\mathrm{d}\boldsymbol{y}}(\boldsymbol{x} - \boldsymbol{y})$

$$\nabla_{\text{1st-order}} := \sum_{\boldsymbol{x} \in \{0,1\}_k^n} \sum_{\boldsymbol{y} \in \{0,1\}_k^n} p(\boldsymbol{y}) \frac{\mathrm{d}f(\boldsymbol{y})}{\mathrm{d}\boldsymbol{y}}(\boldsymbol{x} - \boldsymbol{y}) \frac{\mathrm{d}p(\boldsymbol{x})}{\mathrm{d}\boldsymbol{\theta}}$$

$$= \sum_{\boldsymbol{y} \in \{0,1\}_k^n} p(\boldsymbol{y}) \frac{\mathrm{d}f(\boldsymbol{y})}{\mathrm{d}\boldsymbol{y}} \underbrace{\sum_{\boldsymbol{x} \in \{0,1\}_k^n} \boldsymbol{x} \frac{\mathrm{d}p(\boldsymbol{x})}{\mathrm{d}\boldsymbol{\theta}}}_{=\frac{\mathrm{d}\mathbb{E}[\boldsymbol{x}]}{\mathrm{d}\boldsymbol{\theta}}} - \underbrace{\sum_{\boldsymbol{y} \in \{0,1\}_k^n} p(\boldsymbol{y}) \frac{\mathrm{d}f(\boldsymbol{y})}{\mathrm{d}\boldsymbol{y}} \boldsymbol{y} \sum_{\boldsymbol{x} \in \{0,1\}_k^n} \frac{\mathrm{d}p(\boldsymbol{x})}{\mathrm{d}\boldsymbol{\theta}}}_{=0}$$

$$= \sum_{\boldsymbol{y} \in \{0,1\}_k^n} p(\boldsymbol{y}) \frac{\mathrm{d}f(\boldsymbol{y})}{\mathrm{d}\boldsymbol{y}} \frac{\mathrm{d}\mathbb{E}[\boldsymbol{x}]}{\mathrm{d}\boldsymbol{\theta}} = \mathbb{E}\left[\frac{\mathrm{d}f(\boldsymbol{x})}{\mathrm{d}\boldsymbol{x}} \frac{\mathrm{d}\mathbb{E}[\boldsymbol{x}]}{\mathrm{d}\boldsymbol{\theta}}\right]$$

Which proves

$$\mathbb{E}[\nabla_{\text{ST}}] = \nabla_{\text{1st-order}}$$

$\square$

---

**Algorithm 4** ReinMax (Liu et al., 2023)

---

**Require:** $\boldsymbol{\theta} \in \mathbb{R}^n, \tau \in \mathbb{R}_+$
1: $\boldsymbol{\pi}_0 \leftarrow \text{softmax}(\boldsymbol{\theta})$
2: $\boldsymbol{x} \sim \text{Categorical}(\boldsymbol{\pi}_0)$                                                      $\triangleright \mathbb{E}[\boldsymbol{x}] = \boldsymbol{\pi}_0$
3: $\boldsymbol{\pi}_1 \leftarrow \frac{1}{2}(\boldsymbol{x} + \text{softmax}(\boldsymbol{\theta}/\tau))$
4: $\boldsymbol{\pi}_1 \leftarrow \text{softmax}(\text{stop gradient}(\log(\boldsymbol{\pi}_1) - \boldsymbol{\theta}) + \boldsymbol{\theta})$     $\triangleright \log(\text{softmax}(\boldsymbol{x})) = \boldsymbol{x} + c, \forall c \in \mathbb{R}$
5: $\boldsymbol{\pi}_2 \leftarrow 2\boldsymbol{\pi}_1 - \frac{1}{2}\boldsymbol{\pi}_0$
6: $\boldsymbol{x} \leftarrow \text{stop gradient}(\boldsymbol{x} - \boldsymbol{\pi}_2) + \boldsymbol{\pi}_2$
7: **return** $\boldsymbol{x}$

---

**Algorithm 5** Second-order estimator for top-$k$ sampling

---

**Require:** $\boldsymbol{\theta} \in \mathbb{R}^n, k \in (0, n), \tau \in \mathbb{R}_+$
1: $\boldsymbol{\pi}_0 \leftarrow \sigma_k(\boldsymbol{\theta})$
2: $\boldsymbol{x} \sim \pi\text{ps sampling}(\boldsymbol{\pi}_0)$                 $\triangleright$ With exact inclusion probabilities, so that $\mathbb{E}[\boldsymbol{x}] = \boldsymbol{\pi}_0$
3: $\boldsymbol{\pi}_1 \leftarrow \frac{1}{2}(\boldsymbol{x} + \sigma_k(\boldsymbol{\theta}/\tau))$
4: $\boldsymbol{\pi}_1 \leftarrow \sigma_k(\text{stop gradient}(\sigma^{-1}(\boldsymbol{\pi}_1) - \boldsymbol{\theta}) + \boldsymbol{\theta})$         $\triangleright \sigma^{-1}(\sigma_k(\boldsymbol{x})) = \boldsymbol{x} + c, \forall c \in \mathbb{R}$
5: $\boldsymbol{\pi}_2 \leftarrow 2\boldsymbol{\pi}_1 - \frac{1}{2}\boldsymbol{\pi}_0$
6: $\boldsymbol{x} \leftarrow \text{stop gradient}(\boldsymbol{x} - \boldsymbol{\pi}_2) + \boldsymbol{\pi}_2$
7: **return** $\boldsymbol{x}$

---

**A potential second-order estimator.** ReinMax (Liu et al., 2023) is a second-order generalization of the straight-through estimator. It is derived using two properties of softmax and categorical sampling:

- The inverse of softmax, up to an additive constant, is $\log$.
- The expectation of categorical sampling is its parameter.

We derive a potentially second-order estimator for top-$k$ sampling. With sigmoid top-$k$ and $\pi$ps sampling, we have the following:

- The inverse of sigmoid top-$k$, up to an additive constant, is $\sigma^{-1}$ (Proposition 1).
- The expectation of $\pi$ps sampling with exact inclusion probabilities is its parameter.

We test this estimator on a generalized version of the polynomial programming problem in Liu et al. (2023).

$$\min_{\boldsymbol{\theta}} \mathbb{E}_{\boldsymbol{x} \sim \pi\text{ps sampling}(\sigma_k(\boldsymbol{\theta}))} \left[ \|\boldsymbol{x} - \boldsymbol{c}\|^2 \right]$$

with $n = 100$, $k = 50$ ($n = 2$ and $k = 1$ in the original paper) and $\boldsymbol{c}$ having $k$ dimensions equal to 0.45 and the rest 0.55. 128 problems are solved in parallel, and we draw batches of 256 samples for each step. We observe the same improvement compared to straight-through estimation as in Liu et al. (2023), see Figure 9.

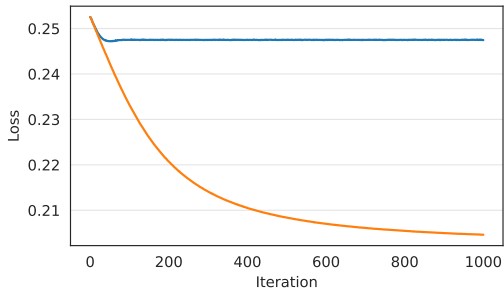

Figure 9: **Polynomial programming.** The second-order estimator (orange) converges to the optimum, while straight-through (blue) does not.

## D  EXPERIMENTAL DETAILS

**Hardware.** Experiments were run using a single GPU, either an NVIDIA RTX 2080 Ti or an NVIDIA A40. The wall-clock times reported in Figure 6 and Appendix A were recorded using an NVIDIA RTX 2080 Ti with 12 GB of VRAM.

**Network architectures.** For both the feature selection and VAE experiments, we use dense ReLU networks with two hidden layers of size 512 and 256 (in reversed order for the decoder). The decoder's output is passed through a sigmoid, and we use binary cross-entropy as the reconstruction loss.

**Hyperparameters.** We use the Adam optimizer (Kingma & Ba, 2015) with default parameters ($\beta_1 = 0.9$ and $\beta_2 = 0.999$) and no weight decay. We use a learning rate of $10^{-3}$ for feature selection and $10^{-4}$ for the VAE. We let all temperatures $\tau = 1$.

## E  TOP-$k$ DIAGRAM

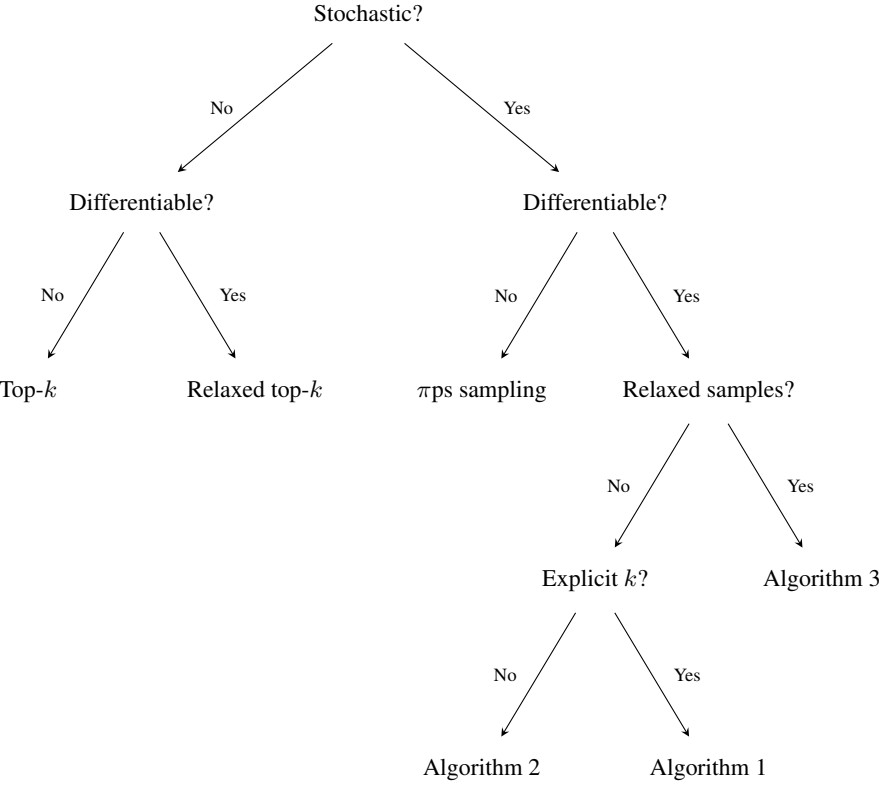

Figure 10: **Top-$k$ diagram.** A diagram categorizing the different notions of top-$k$ in this work. From left to right, the different settings and their corresponding sections are top-$k$ (§1), relaxed top-$k$ (§2.1), $\pi$ps sampling (§2.2), sampling with explicit (Sections 2.1 and 2.2) and implicit $k$ (§2.3), and relaxed sampling (§2.5). Gradient estimates for the last three are discussed in §2.4.

