# OpenReview forum: "Differentiable Top-k: From One-Hot to k-Hot"
_ICLR.cc/2026/Conference — Submitted to ICLR 2026_

### Official Review · Reviewer_givJ · 2025-10-23

**Soundness:** 2
**Presentation:** 1
**Contribution:** 2
**Rating:** 4
**Confidence:** 2

**Summary:**

The paper introduces a new differentiable relaxation of the top-k operator, called sigmoid top-k. The authors derive properties of sigmoid top-k, such as differentiability w.r.t. to $k$ as well as the input $x$ and show that sigmoid top-k leads to an entropy-regularized optimization problem. The paper then discusses different top-k sampling strategies. Finally, the authors show that their proposed method has better scalability than previous approaches and slightly improved performance (w.r.t. test loss).

**Strengths:**

* S1: The proposed method seems to noticeably improve in terms of scalability compared to previous methods.

* S2: The proposed method allows for continuous values of $k$ that can even be optimized for. When optimizing for $k$, the method seems to robustly converge to an optimal value of $k$ (Fig. 4).

* S3: The authors show that sigmoid top-k is equivalent to solving a binary-entropy-regularized optimization problem (Proposition 2).

**Weaknesses:**

* W1: The paper is not well presented, e.g., it lacks motivation. Why is it interesting to study relaxations of top-k? Why do we even need to generalize from one-hot to k-hot? For example, discussing related work earlier (e.g., in introduction or as section 2) could motivate this from a scalability perspective.

* W2: Given the very simple datasets used (i.e., MNIST and Fashion-MNIST), the experimental setup does not seem convincing to me. While the demonstration of scalability (Figure 5) appears valid, the performance improvements in Tables 2 and 3 seem negligible and sometimes even inconsistent. The results and their potential significance are also not discussed or contextualized in the text.

* W3: It'd be meaningful to add experiments comparing the shifted sigmoid variants due to the similarity to the proposed approach.

## Comments

* C1: Figures 4 and 5 are quite misplaced from their text (e.g., Figure 4 is on page 6 and only referenced on page 9)

* C2: The font size of table and figure captions does not match the text font size. The authors should refrain from changing font sizes from the official style file.

**Questions:**

* Q1: I assume that the VAEs from Table 3 were trained on image reconstruction. If so, can the authors provide qualitative comparisons in addition to the test losses?

---

> ### Author Response · Authors · 2025-11-23
>
> Thank you for your thoughtful review! Please see our response below.
>
> **W1 Presentation and motivation**
>
> *Why is it interesting to study relaxations of top-k?*
>
> As mentioned in the paragraph "Differentiable relaxations" (Section 1), the non-differentiability of top-$k$ hinders gradient-based optimization. Furthermore, the paragraph "Applications" (also Section 1) lists multiple applications of $k$-hot vectors.
>
> *Why do we even need to generalize from one-hot to k-hot?*
>
> Generalizing from one-hot to $k$-hot is a useful approach for designing $k$-hot relaxations and sampling. In the paper, we make sure certain properties from the one-hot case are carried over to the $k$-hot case. This is what allows us to prove that straight-through estimation is a first-order gradient approximation for Algorithm 1 (see Proposition 3). Moreover, it leads us to a modular approach that decomposes the problem into two subproblems: relaxed top-$k$ and $\pi$ps sampling.
>
> *Moving the related works section*
>
> Our method starts from the one-hot case and generalizes each of its components, so we intentionally placed the related works after the method, since it does not directly build upon previous approaches.
>
> Presentation is important, so please respond with any additional questions or suggestions that can help improve the paper.
>
> **W2 Experiments**
>
> See the general comment regarding the datasets. Regarding the improvements, we do not expect Algorithm 1 to outperform SIMPLE. As discussed in Section 3, SIMPLE essentially implements the same gradient estimate.
>
> **W3 Shifted sigmoid variants**
>
> We assume that this refers to the shifted sigmoid variants mentioned in Section 3. Unlike sigmoid top-$k$, these shifted sigmoids do not project onto the $k$-capped simplex. The values will be in $[0, 1]^n$ but not sum to $k$, so they play a slightly different role than sigmoid top-$k$.
>
> **Q1 VAE reconstructions**
>
> Yes, this is correct! We will try to add these if time allows. Our experience has been that the visual quality is well-aligned with MSE for these reconstructions.

---

### Official Review · Reviewer_ExNU · 2025-10-23

**Soundness:** 2
**Presentation:** 4
**Contribution:** 2
**Rating:** 4
**Confidence:** 2

**Summary:**

This paper develops a framework (sigmoid top-k) for differentiable top-k sampling operations, generalizing the one-hot case to k-hot cases. The sigmoid top-k framework is easy to compute and show efficiency advantage over prior work (SOFT). Moreover, the authors generalize $\pi ps$ sampling from one-hot to k-hot, and discuss how to implicitly learn the k. Empirically, this paper validates the proposed methods on feature selection and sparse representation learning, showing lower loss and faster runtime than prior differentiable top-k relaxations on MNIST and Fashion MNIST.

**Strengths:**

**Unified theoretical framework** This paper connects differentiable relaxations (softmax to sigmoid top-k) and differentiable sampling (categorical to $\pi ps$) under one consistent view, which provides conceptual clarity that was missing in prior work. Moreover, the authors shows that the output of sigmoid top-k can be understood as a regularized optimization problem (see Proposition 2) which provides mathematical explanation of the algorithms.

**Novel Top-k algorithm** The authors propose an efficient scalable differentiable top-k algorithm, and demonstrate its efficiency advantage over other sampling algorithms (see Figure 5). Moreover, the authors propose using phantom unit mechanism to handle the emergent non-integer k during the learning process. Finally, the algorithm replies a simple root solver which is easy to implement and memory-efficient, enabling applications in high-dimensional problems.

**Clear presentation** The paper is well-written, with comprehensive properties and illustrative examples of the top-k sampling.

**Weaknesses:**

**Limited theoretical novelty** Though this paper offers an elegant and unified framework connecting the k-hot sampling with differentiable relaxations, much of its contribution builds upon prior works. The proposed sigmoid top-k framework can be viewed as a natural extension of softmax when constraining $\sum p_i=k$ instead of $\sum p_i=1$. Moreover, as is pointed out in the paper, entropy-regularized optimization problem and k-capped simplex have been studied in prior work. Therefore, I think the novelty of this paper lies in the combination of these existing ideas rather than discovery of them.

**Lack of deeper analysis of optimization behavior** This paper lacks discussion of how the error of root solver affects gradient accuracy, convergence stability, or numerical conditioning in practical optimization. Similarly, while the authors prove that the straight-through estimator yields a first-order approximation of the true gradient for top-k sampling (see Proposition 3), there is no analysis of bias, variance,  and comparison with existing estimators.

**Limited empirical evaluation** The experimental section demonstrates clear improvements on two relatively simple tasks: feature selection and sparse representation learning on MNIST and Fashion MNIST. Though promising, the tasks seem too simple to convince me that this method does work in more general tasks.

**Missing interpretation in the context of probabilistic modeling** While the paper discusses the $\pi ps$ sampling design, it lacks the discussion of how the proposed algorithm will affect the probabilistic modeling. For example, does the sigmoid top-k lead to a normalized probability model over subsets?

**Questions:**

**Question 1** Does the algorithm applies to beam search? I'm thinking of the application of sampling in LLMs where one needs to sample a sequence of tokens. If the algorithm can demonstrate superior performance when applied to beam search, this can be a huge plus.

**Question 2** How does the algorithm work in the multi-label classification problems?

**Question 3** What is the trade-offs between the entropy regularization strength and approximation accuracy of the true gradient? Can you rigorously characterize it?

---

> ### Author Response · Authors · 2025-11-23
>
> Thank you for your detailed review! We are delighted to see the conceptual clarity compared to prior work and sigmoid top-$k$ among the paper's strengths. Please see our answers to your questions and concerns below.
>
> **W1 Limited theoretical novelty**
>
> Our proposed method absolutely builds upon existing works. This is a key strength of our method. Instead of treating differentiable top-k sampling as an irreducible problem, we decompose it into 1) relaxed top-k and 2) πps sampling. Sigmoid top-$k$ is a novel approach to relaxed top-$k$. Practically, it is very simple to implement, while sharing some of the theoretical. While it can be seen as a generalized softmax, "a natural extension of softmax when constraining $\sum p_i = k$ instead of $\sum p_i = 1$" is not quite accurate. The problem of finding such a relaxed top-$k$ function has been the topic of multiple papers, such as [1, 2].
>
> **W2 Lack of deeper analysis of optimization behavior**
>
> First, let us address the root-finding and implicit differentiation. Blondel et al. [3] demonstrate that the Jacobian error is a function of the solution error, that is, the accuracy of sigmoid top-k’s gradients depends on the accuracy of the root-finding solution (this is intuitive, as the implicit gradient is exact at the solution). Luckily, we can solve this problem rapidly with guaranteed convergence, as discussed in Appendix A, root finding. Empirically, we are able to pass `torch.autograd.gradcheck` tests, as shown in the tutorial notebook.
>
> Second, the straight-through estimator. It is a biased estimator. As stated in Section 3, SIMPLE implements the same estimator and is hence
>
> **W3 Limited empirical evaluation**
>
> See the general comment.
>
> **W4 Missing interpretation in the context of probabilistic modeling**
>
> Sigmoid top-k takes a real-valued input and produces parameters (inclusion probabilities) for a $\pi$ps sampling design. In probabilistic modeling, $\pi$ps sampling designs correspond to different distributions over the set of $k$-hot vectors. Depending on the specific design, these may have known probability mass functions that are normalized over all $k$-hot vectors (see, e.g., Tillés' book [4]). For example, conditional Poisson sampling has the pmf:
> $$p(\boldsymbol{x}) \propto \prod_{i=1}^n p_i^{x_i} (1 - p_i)^{(1 - x_i)}$$
> where $p_i$ are the desired inclusion probabilities.
>
> **Q1 Beam search**
>
> In principle, yes! As shown in previous works, both top-k sampling [5] and top-$k$ relaxations [1] can be used in beam search.
>
> **Q2 Multi-label classification**
>
> Differentiable top-$k$ is useful in a particular type of multi-label classification problems. In the usual setting, you would use independent sigmoid activations. At inference time, predictions are obtained by applying a threshold to the sigmoid outputs. This makes sense, as the sigmoid is a differentiable relaxation of the threshold. If we are interested in sampling predictions, we can use the sigmoid outputs as Bernoulli parameters.
>
> Now, if the number of positive labels is known, the task reduces to predicting the top-$k$ most likely labels. Modeling this with independent sigmoid activations risks predicting an incorrect number of labels. Instead, we can replace the standard sigmoids with sigmoid top-k. At inference time, we predict the top-$k$ labels. Again, this makes sense as sigmoid top-$k$ is a differentiable relaxation of top-$k$. For sampling predictions, we can use the sigmoid top-$k$ outputs as πps sampling parameters.
>
> **Q3 Entropy regularization**
>
> This is an interesting question. In the paper, we derive an equivalence between sigmoid top-$k$ and an entropy-regularized optimization problem (Proposition 2). Changing the weight of this regularization is, in fact, equivalent to temperature scaling (Proposition 1). We have added Corollary 1 to state this fact in the paper. From this temperature perspective, we have an asymptotic understanding of bias and variance. Higher temperature leads to higher bias and lower variance, and vice versa.
>
> **References**
>
> [1] Differentiable Top-k with Optimal Transport, Xie et al., 2020.
>
> [2] Fast, Differentiable and Sparse Top-k: a Convex Analysis Perspective, Sander et al., 2023.
>
> [3] Efficient and Modular Implicit Differentiation, Blondel et al., 2022.
>
> [4] Sampling Algorithms, Tillé, 2006.
>
> [5] Stochastic Beams and Where To Find Them: The Gumbel-Top-k Trick for Sampling Sequences Without Replacement, Kool et al., 2019.

---

> > ### Comment · Reviewer_ExNU · 2025-11-25
> >
> > Thank you very much for the responses and you have addressed most of my concerns. My remaining question is about the practical usefulness of the proposed framework. Although the authors note in the general response that their experiments are already larger in scale compared to prior work, the experimental setup is still limited to a VAE on MNIST. As a result, it is difficult for me to assess how well the method would extend to more relevant and widely studied settings, such as sampling in modern large-scale generative models.

---

> > > ### Author Response · Authors · 2025-11-26
> > > **Beam search**
> > >
> > > Thank you for your timely response. We are happy to have addressed most of your concerns. If we understand you correctly, the remaining concern is applicability to modern models.
> > >
> > > As a proof-of-concept, we have modified some existing code for beam search ([URL](https://github.com/mikecvet/beam)) using the Hugging Face T5 model to use sigmoid top-$k$ instead of softmax. We only had to replace a few of lines of code. In beam search, $n$ is the number of tokens, and $k$ is the beam width. Even in this small example, $n = 32128$ and $k = 4$, so a scalable top-$k$ relaxation like sigmoid top-$k$ is necessary. Prior works, such as SOFT top-$k$ may not be computationally feasible at this scale, especially in state-of-the-art models where the number of tokens is even larger (see Figure 6).
> > >
> > > Let us look at a snapshot of the search to see why sigmoid top-$k$ may be preferable to softmax. The current sequence is:
> > >
> > > `"the quick brown fox jumps over the lazy dog" is an English-language pangram. it[token]`
> > >
> > > | p(`[token]`)    | Raw logit | Softmax ($\tau=0.5$) | Sigmoid top-$k$ ($\tau=0.5$) | Softmax ($\tau = 5.0$) |
> > > |--------------|------------------|------------------|-------------------------|----------------|
> > > | `▁contains`  | -1.47  | 0.56628484       | 0.99054915              | 0.00117765     |
> > > | `▁is`          | -1.62 |0.41698441       | 0.98720860              | 0.00114216     |
> > > | `▁uses`        | -3.82 | **0.00515623**       | **0.48831841**              | 0.00073612     |
> > > | `▁was`         | -3.94 | **0.00402238**       | **0.42676386**              | 0.00071806     |
> > > |Next tokens after the top-4 |||
> > > | `▁has`         | -4.22 | 0.00230794       | 0.29931045              | 0.00067926     |
> > > | `▁includes`    | -4.38 | 0.00165868       | 0.23488662              | 0.00065719     |
> > > | `'`             | -4.48 | 0.00137800       | 0.20321719              | 0.00064512     |
> > > | `▁`            | -4.74 | 0.00080090       | 0.12909850              | 0.00061104     |
> > >
> > > Softmax results approaches argmax at low tempteratures. In this example, only two of the top-$k$ selections has a meaningful score. Both `▁uses` and `▁was` are assigned around 100 times lower scores than the top-2, even though we are searching for the top-4. If you decrease the temperature, this difference **grows** as softmax approaches argmax.
> > >
> > > Sigmoid top-$k$, on the other hand, approaches top-$k$ at low temperatures. We see that all of the top-$k$ options have a meaningful score. `▁uses` and `▁was` are only around half of the scores in the top-2. If you decrease the temperature, this difference **shrinks** as sigmoid top-$k$ approaches top-$k$. In addition, these scores can be used as inclusion probabilities for stochastic beam search with $\pi$ps sampling, like in Algorithm 1.
> > >
> > > To achieve a similar effect with softmax, we could (naively) increase the temperature. However, this flattens the distribution over **all** 32128 tokens. This is fine if we are picking the top-$k$ options deterministically, but would cause excessive randomness in stochastic beam search.
> > >
> > > Asymptotically, we have:
> > >
> > > | p(`[token]`)    | Raw logit | Softmax ($\tau \rightarrow 0$) | Sigmoid top-$k$ ($\tau \rightarrow 0$) | Softmax ($\tau \rightarrow \infty$) |
> > > |--------------|------------------|------------------|-------------------------|----------------|
> > > | `▁contains`  | -1.47  | 1       | 1              | 1 / 32128     |
> > > | `▁is`          | -1.62 |0       | 1             | 1 / 32128     |
> > > | `▁uses`        | -3.82 | 0       | 1              | 1 / 32128     |
> > > | `▁was`         | -3.94 | 0       | 1              | 1 / 32128    |
> > > |Next tokens after the top-4 |||
> > > | `▁has`         | -4.22 | 0       | 0              | 1 / 32128     |
> > > | `▁includes`    | -4.38 | 0       | 0              | 1 / 32128 |
> > > | `'`             | -4.48 | 0       | 0              | 1 / 32128    |
> > > | `▁`            | -4.74 | 0       | 0              | 1 / 32128     |
> > >
> > >
> > > In essence, **softmax assumes $k = 1$, while sigmoid top-$k$ is informed about $k$**.
> > >
> > > Apart from inference, our methods could, in principle, be used in deterministic or stochastic beam search during training, since they are differentiable.

---

### Official Review · Reviewer_77HM · 2025-10-28

**Soundness:** 3
**Presentation:** 2
**Contribution:** 2
**Rating:** 4
**Confidence:** 2

**Summary:**

The authors propose a novel differentiable relaxation to the top-K operator as the sigmoid-k function. They define the operator for continuous k, handling implicit values of k in the process. They show that this operator can be combined with stochastic relaxation techniques similar to gumbel-softmax and straight through estimator. Empirically the method is evaluated on MNIST and Fashion-MNIST.

**Strengths:**

1. The method is technically sound and seems simple to implement.
2. The algorithm models the inclusion probabilities exactly.

**Weaknesses:**

1. Exposition of the method and prior work can be improved.  Section 2, I believe is poorly organized for a novice reader with limited background. $\pi$ps sampling (which I believe is an R package and not an algorithm?) is a bit hard to understand what it exactly is, even after multiple reads of the paper.
2. The limited evaluation setup might not convince the reader of the utility of the method proposed. I believe prior works have shown the superiority of their categorical modeling on more challenging and recent tasks like stochastic beam search. I am not sure if something like that would be possible.
3. A clear (empirical) comparison with the baseline methods considered (and prior work) would help the appreciate the novelty and technical challenges better. There are a lot of baselines which are considered in the experiment section like SIMPLE, Soft Top-k (among others). A baseline subsection clearly elaborating and explaining the baselines would help the paper.

**Questions:**

1. Should the conditions in Eqn 15 be switched? (Just clarifying, I might have every well mis-understood the equation or made error in my calculations)

---

> ### Author Response · Authors · 2025-11-23
>
> Thank you for your thoughtful feedback! We are pleased to hear that you find our method novel and technically sound. Please see our answers to your review below.
>
> **W1 Method and prior work**
>
> $\pi$ps sampling is neither an R package nor an algorithm. It is short for “probability proportional to size sampling without replacement” (as opposed to pps sampling, see this [URL](https://en.wikipedia.org/wiki/Probability-proportional-to-size_sampling), which is done with replacement). The terminology comes from the field of sampling design and survey sampling (see, e.g., Tillé’s book [1] and the references therein). In other words, it is the problem of sampling a $k$-hot vector with given marginal probabilities. As explained in the paper, many distributions (or sampling designs) share the same marginals, which is why there are many possible designs for $\pi$ps sampling. We understand that this terminology is not established in machine learning, so we are keen to explain it clearly. If possible, please elaborate
>
> **W2 Experiments**
>
> See the general comment. In principle, the method could be used for beam search.
>
> **W3 Baseline comparison**
>
> We have added Table 2 to give an overview of recent approaches to relaxed top-$k$. We have also added Figure 7 to illustrate the main baselines considered in the experiments. The latter should make the limitations of previous methods clear and highlight the simplicity of our approach.
>
> **Q1 Equation 15**
>
> You are correct, thank you for spotting this! The case when the phantom unit is selected results in $k = \lfloor \sum_{i=1}^n \pi_i \rfloor$, and when it is not selected, $k = 1 + \lfloor \sum_{i=1}^n \pi_i \rfloor$. We have updated this in the revised paper. Again, thank you for this.
>
> **References**
>
> [1] Sampling Algorithms, Tillé, 2006.

---

### Official Review · Reviewer_rWG5 · 2025-11-03

**Soundness:** 2
**Presentation:** 2
**Contribution:** 3
**Rating:** 2
**Confidence:** 5

**Summary:**

Nice work in sense of technology development and extending stochastic/relaxed discrete sampling. However, the main problem are 1)not enough large scale exerperiment 2) not good writting

The writting and experiments looks to me potentially purely generated by Chatgpt /claude code, which I could be wrong. I request to see the code.

**Strengths:**

discrete representation optimization is an important problem in machine learning , such as sparse latent, quantization etc. Traditionally, there are in general 3 ways to walk around the non-differnetiability problem 1) fake it (eg. STE) 2) relax it (gumbel softmax) 3) shortcut it (eg. REINFORCE, each with its own problem

Most research in the field focus on one hot representation, and relatively little has been done on top-k. This makes contribution by this manuscript unique.To me personally, two major contribution by this manuscript are 1) the sigmoid top k which serves as a relaxation ( in a similar style as Hinton sigmoid straight through) and 2) the implicit "top k" ( which I think the name is misleading)

**Weaknesses:**

The few weakness:

1) the introduction part is not well written and difficult to follow, looks very likey a LLM generated piece of text

2) value of implicit top k and application is unclear.

3) The author mentioned mixture of expert, ,however most recent works on MOE are no longer using this style of discrete choice now. Correct me if I am wrong.

4) most importantly: the experiments are too simple and to small scale. A large scale experiment on modern architecture are definitely needed

**Questions:**

I am curious about formulation in equation 9.  how about the bias and variance your relaxation? Also, shared by all relaxation approach, how is the sampling distribution( during inference) distribution from model trained using your method different from model trained using other estimation method

If the author can 1) show that the manuscript and code are not LLM generated 2) explain behavior of their relaxation method 3) provide at least one solid large scale experiment , I am happy to consider raise the score to >= 6

---

> ### Author Response · Authors · 2025-11-23
>
> Thank you for your comments! We have tried to address your concerns below. Please consider raising your score if these answers are satisfactory, and if not, please let us know what doubts remain.
>
> **W1 LLM usage and code**
>
> We are puzzled by the claim that the writing and experiments are AI-generated. LLMs were used in accordance with the ICLR 2025 guidelines. We reiterate the section on LLM usage included in the paper:
>
> > LLMs were used to give feedback on writing, including rephrasing sentences, and code for figures and equations. For coding, LLMs were used to generate starting points, autocompletion, and code for plots.
>
> We support our case with three main points:
> 1. For the text, which was written in Overleaf, we have edit history that shows it being written manually and incrementally (evidence can be provided to the AC/PCs if necessary).
> 2. See this [URL](https://iclr.pangram.com), a LLM-based evaluation of ICLR submissions, where the paper is listed as **0% AI-generated** (only 61% of all submissions are within the 0-10% range).
> 3. Regarding code, we have added the code for the main experiments to the anonymous repo, which is already linked in the reproducibility statement ([URL](https://anonymous.4open.science/r/diff-topk/)).
>
> Finally, we appreciate the reviewer’s concern. However, it should be noted that this is a serious claim, and the burden of proof for illegitimate LLM use rests on the reviewer’s side. In that respect, the reviewer has not offered any evidence to support this claim in their initial review.
>
> **W2 Implicit k**
>
> First, let us clarify what implicit $k$ entails, which also explains why the naming is not misleading. The key difference between Algorithm 1 and Algorithm 2 is that k is given as an input to Algorithm 1 and not to Algorithm 2. In Algorithm 1, we use sigmoid top-$k$ to explicitly enforce a given $k$, while the formula $k = \sum_{i=1}^n \sigma(x_i)$, in Algorithm 2, determines the implicit value of $k$. This way, it is implicitly determined from $x$ instead of being an explicit parameter to the algorithm. Using Implicit $k$ (Section 2.3 and Algorithm 2) has multiple use-cases:
> - It eliminates the need for a relaxed top-$k$ function. In Algorithm 2, we simply use the standard sigmoid, which is simpler and faster than top-$k$ relaxations.
> - It makes learning $k$ easy, since it is a function of the input, which is already learned. This is useful in situations where you want to control $k$ through a loss function (see Figure 5).
>
> **W3 Mixture of Experts**
>
> Mixture of expert (MoE) techniques are developing rapidly, and we do not know if top-$k$ routing techniques will be favored going forward, but they have definitely been a popular approach in recent years. Two key challenges in MoE routing are 1) differentiability and 2) exploration. We think sigmoid top-$k$ and our approach to top-$k$ sampling could be used to address both of these problems. A common approach for MoE, used in e.g., DeepSeek [1], is `topk(softmax(x))`. Here, top-$k$ and softmax are mismatched in the sense that top-$k$’s codomain $\\{0,1\\}_k^n$ is not a subset of softmax’s codomain $\Delta^{n-1}$ for $k \neq 1$. Using `topk(sigmoid_topk(x))` instead would fix this mismatch. That being said, MoE is just one of many possible applications (see Section 1, applications).
>
> **W4 Experiments**
>
> See the general comment.
>
> **Q1 Relaxation**
>
> *Bias and variance*
>
> Equation 9 defines sigmoid top-$k$, our proposed differentiable top-$k$ relaxation. It shares many properties with softmax (see Proposition 1). As stated in Proposition 1 and illustrated in Figure 3, sigmoid top-$k$ can be tempered like softmax. As mentioned in the paper, the temperature can be used to control the bias-variance trade-off just like you would for softmax.
>
> *Training vs. inference*
>
> In the paper, we propose multiple ways this relaxation can be used for differentiable top-$k$ (see Appendix E for an overview). Let us reiterate three ways it can be used here:
> 1. As a deterministic relaxation of top-$k$. The same way you would use softmax.
> 2. For hard, discrete sampling using $\pi$ps sampling (Algorithm 1).
> 3. For relaxed sampling with Gumbel noise (Algorithm 3).
>
> Just like for softmax, there may be a difference between training and inference time, depending on the use case. For 1, there is no difference if sigmoid top-$k$ is used during inference. For 2, there is no difference since hard samples are used during training. For 3, there is a difference if relaxed samples are used during training (but not for the straight-through version, which we used in our experiments).
>
> **References**
>
> [1] DeepSeekMoE: Towards Ultimate Expert Specialization in Mixture-of-Experts Language Models, Dai et al., 2024.

---

### Author Response · Authors · 2025-11-23

We would like to thank all reviewers for their feedback. A majority of reviewers agree that the method is mathematically sound and computationally efficient. In this comment, we provide information and answers that are relevant to all reviewers. Please do not hesitate to **ask follow-up questions** during the remainder of the discussion period, and consider **raising your scores** if we have addressed your concerns.

**Tutorial notebook**

In the anonymous repository linked in the reproducibility statement ([URL](https://anonymous.4open.science/r/diff-topk/)), we have included a tutorial notebook with a self-contained implementation of the paper's core algorithms. We recommend reading through it to see how the method is implemented in practice. We have also added the code used for the experiments as requested by Reviewer rWG5.

**Experiments**

A common concern among reviewers is the size of the experiments. We agree that MNIST is a small dataset, but the paper’s objective and contributions are not necessarily dependent on experiments with large models or datasets:
1. This work was intended to be mainly a theoretical study. We go back to first principles and derive a formally justified approach to differentiable top-$k$. Our method has proven advantages compared to existing state-of-the-art methods, namely: a modular approach similar to the one-hot case, sampling with exact inclusion probabilities and $k$, and significantly improved scalability. The goal of the provided experiments is twofold: to demonstrate a practical implementation and to establish the scalability of the proposed approach in practice.
2. We emphasize that the **problem size** is different from the **dataset size**. The dimensionality $n$ and subset size $k$ are the problem size for top-$k$. In fact, both our VAE experiments and runtime comparisons demonstrate scalability where we use $n = 1000$ and $k = 500$, which is considerably beyond experiments in SIMPLE [1] and SFESS [2], published in ICLR'23 and ICLR'25, respectively. In their experiments, $n=10$ and $k=5$ were used.
3. The feature selection and VAE experiments have been used as benchmarks in and were inspired by previous works on differentiable $k$-hot sampling [1, 2].

This being said, the applications mentioned by reviewers are very interesting, and we hope that the researchers working on said problems will experiment with our method after publication.

**Changes in the revised version**

- Added Corollary 1 as an answer to Reviewer ExNU’s question.
- Changed the notation for inclusion probabilities from $p$ to $\pi$.
- Swapped the conditions in Equation 15 that were in the wrong order, as pointed out by Reviewer 77HM.
- Added Table 2 to give an overview of top-$k$ relaxations.
- Added Figure 1 and Figure 7 to give a graphical overview of Algorithms 1-3 and clarify their relation to the baselines.
- Added main experiments to the repository ([URL](https://anonymous.4open.science/r/diff-topk/)) as requested by Reviewer rWG5.

**References**

[1] SIMPLE: A Gradient Estimator for k-Subset Sampling, Ahmed et al., 2023.

[2] SFESS: Score Function Estimators for k-Subset Sampling, Wijk et al., 2025.

---

### Meta-Review · Area_Chair_eJSL · 2026-01-06

**Summary:**

The reviews on this paper are borderline -- in general the reviewers liked the core idea of sigmoid top-k, and its simplicity and scalability. It is clear that differentiable top-k methods are interesting to study and relevant to the community.

However, the reviewers generally thought the quality of presentation/clarity of contributions was low. They also wanted to see more extensive experimental results, on more relevant applications. Given the simplicity of the core theoretical idea, I agree that to pass the bar for publication, the idea should be backed up by stronger empirical evidence of performance/applicability.

Note: One reviewer thought that the paper may be LLM generated. I agree with the authors that there is not sufficient evidence for this and am ignoring the point in my metareview.

**Reviewer Concerns:**

Many reviewers wanted to see experiments in more modern/relevant contexts -- e.g. stochastic beam search for text generation. The authors provided some initial evidence that their method could be applied in this context, but did not provide significant new experimental results.

Many reviewers were concerned with the clarity of the paper and the quality of the writing. These concerns still seem to be outstanding. In their rebuttal, the authors mostly argue that the paper is already clear, rather than make an effort to understand why things were confusing for so many of the reviewers and consider editing the writing.

The authors did a nice job of addressing many of the questions/more minore concerns of the reviewers, but I think many of the major concerns still remain.

**Reviewer Scores:**

ExNU -- perhaps would have increased score by 1.

I do not think any of the other reviewers would have increased their scores.

---

### Decision · Program_Chairs · 2026-01-26

Reject